# Learning Structured Distributions From Untrusted Batches: Faster and Simpler

**Sitan Chen**
MIT
sitanc@mit.edu

**Jerry Li**
MSR
jerrl@microsoft.com

**Ankur Moitra**
MIT
moitra@mit.edu

## Abstract

We revisit the problem of learning from untrusted batches introduced by Qiao and Valiant [QV17]. Recently, Jain and Orlitsky [JO19] gave a simple semidefinite programming approach based on the cut-norm that achieves essentially information-theoretically optimal error in polynomial time. Concurrently, Chen et al. [CLM19] considered a variant of the problem where $\mu$ is assumed to be structured, e.g. log-concave, monotone hazard rate, $t$-modal, etc. In this case, it is possible to achieve the same error with sample complexity *sublinear in* $n$, and they exhibited a quasi-polynomial time algorithm for doing so using Haar wavelets.

In this paper, we find an appealing way to synthesize [JO19] and [CLM19] to give the best of both worlds: an algorithm which runs in polynomial time and can exploit structure in the underlying distribution to achieve sublinear sample complexity. Along the way, we simplify the approach of [JO19] by avoiding the need for SDP rounding and giving a more direct interpretation of it via soft filtering, a powerful recent technique in high-dimensional robust estimation. We validate the usefulness of our algorithms in preliminary experimental evaluations.

## 1 Introduction

In this paper, we consider the problem of *learning structured distributions from untrusted batches*. This is a variant on the problem of *learning from untrusted batches*, as introduced in [QV17]. Here, there is an unknown distribution $\mu$ over $\{1, \ldots, n\}$, and we are given $N$ batches of samples, each of size $k$. A $(1-\epsilon)$-fraction of batches are "good," and consist of $k$ i.i.d. samples from some distribution $\mu_i$ at distance at most $\omega$ from $\mu$ in total variation distance, but an $\epsilon$-fraction of batches are "bad," and can be adversarially corrupted. The goal is to estimate $\mu$ in total variation, equivalently $L_1$.

This problem models a situation where we get batches of data from different users, e.g. in a crowdsourcing application. Each honest user provides a small batch of data, which is by itself insufficient to learn a good model, and moreover, can come from slightly different distributions depending on the user, due to heterogeneity. At the same time, a non-trivial fraction of data can come from malicious users who wish to game our algorithm to their own ends. The high level question is whether we can exploit the batch structure of our data to improve the robustness of our estimator. There are three separate, but equally important, metrics under which we can evaluate our estimator:

**Robustness** How accurately can we estimate $\mu$ in total variation distance?
**Runtime** Are there algorithms that run in polynomial time in all the relevant parameters?
**Sample complexity** How few batches do we need in order to estimate $\mu$?

In the original paper, Qiao and Valiant [QV17] focus primarily on robustness. They give an algorithm for learning general $\mu$ from untrusted batches that uses a polynomial number of batches, and estimates $\mu$ to within $O\left(\omega + \epsilon/\sqrt{k}\right)$ in total variation distance, and they proved that this is the best possible

up to constant factors. However, their estimator runs in time $2^n$. Qiao and Valiant [QV17] also gave an algorithm based on low-rank tensor approximation that needs $n^k$ time and batches.

A natural question is whether or not this robustness can be achieved *efficiently*. Chen et al. [CLM19] gave an $n^{\log^2 1/\epsilon}$ time sum-of-squares algorithm that uses $n^{\log 1/\epsilon}$ batches and estimates $\mu$ to within $O\left(\omega + \frac{\epsilon}{\sqrt{k}}\sqrt{\log 1/\epsilon}\right)$ in $L_1$. Concurrently and independently, Jain and Orlitsky [JO19] gave a polynomial time algorithm based on a much simpler SDP that achieves the same error. Their approach was based on an elegant way to combine approximation algorithms for the cut-norm [AN04] with the filtering approach for robust estimation [DKK+17, SCV18, DKK+18, DKK+19, DHL19].

To some extent, the results of [CLM19, JO19] also address the third consideration, sample complexity. [JO19] uses $N = \widetilde{O}(n/\epsilon^2)$ batches to achieve the above error rate. Even without corruptions, for general $\mu$ any algorithm needs at least $\Omega(n/\epsilon^2)$ batches of size $k$ to learn to $L_1$ error $O(\omega + \epsilon/\sqrt{k})$. Thus, their sample complexity is nearly-optimal unless one makes additional assumptions.

Unfortunately, in many cases, domain size $n$ can be very large, and sample complexity growing strongly with $n$ can render the estimator impractical. However in most applications, we have prior knowledge about the *shape* of $\mu$ that could in principle be used to drastically reduce the sample complexity. For example, if $\mu$ is log-concave, monotone, or multimodal with bounded number of modes, $\mu$ can be approximated by a *piecewise polynomial* function, and sans corruptions, this can be used to reduce the sample complexity to logarithmic in $n$ [CDSS14b]. An appealing aspect of the relaxation in [CLM19] was that it could incorporate shape-constraints, via Haar wavelets, allowing them to achieve sample complexity quasipolynomial in $d$ and $s$, respectively the degree and number of parts in the piecewise polynomial approximation, and quasipolylogarithmic in $n$. Unfortunately, while [JO19] achieves better runtime and sample complexity in the unstructured setting, *a priori* their techniques do not extend to obtain a similar sample complexity under structural assumptions.

This raises a natural question: can we build on [JO19] and [CLM19], to incorporate shape constraints into a simple SDP approach that can achieve nearly-optimal robustness, polynomial runtime, and sample complexity sublinear in $n$? In this paper, we answer this question in the affirmative:

**Theorem 1.1.** *Let $\mu$ be a distribution over $[n]$ which is $\eta$-approximated by an $s$-part piecewise polynomial with degree at most $d$. There is an algorithm which runs in time polynomial in all parameters and estimates $\mu$ to within $O\left(\eta + \omega + \frac{\epsilon}{\sqrt{k}}\sqrt{\log 1/\epsilon}\right)$ in total variation after drawing $N$ $\epsilon$-corrupted batches, each of size $k$, where $N = \widetilde{O}\left((s^2 d^2/\epsilon^2)\cdot \log^3(n)\right)$ is the number of batches.*

Any algorithm for this problem must take at least $\Omega(sd/\epsilon^2)$ batches to achieve error $O(\eta+\omega+\epsilon/\sqrt{k})$, and an interesting open question is whether there is a polynomial time algorithm that achieves these bounds. For robust mean estimation for Gaussians, there is evidence for a $\Omega(\sqrt{\log 1/\epsilon})$ gap between the best possible estimation error and what can be achieved by polynomial time algorithms [DKS17]. It seems plausible that the $\Omega(\sqrt{\log 1/\epsilon})$ gap incurred by our result is unavoidable as well.

## 1.1 High-Level Argument

In this work we show how to unite the filtering of [JO19] with the Haar wavelet technology of [CLM19] to obtain a polynomial-time, sample-efficient algorithm for learning structured distributions from untrusted batches. In this section, we will specialize to the case of $\omega = 0$ for the sake of clarity.

**Learning via Filtering** Given a batch of samples $Y_i = (Y_i^1, ..., Y_i^k)$ from a distribution $\mu$ over $[n]$, the frequency vector $\{\frac{1}{k}\sum_{j=1}^k \mathbb{1}[Y_i^j = a]\}_{a\in[n]}$ is distributed according to the normalized multinomial distribution $\text{Mul}_k(\mu)$ given by $k$ draws from $\mu$ (see Section 1.3 for notation). Note that $\mu$ is precisely the mean of $\text{Mul}_k(\mu)$, so the problem of estimating $\mu$ from an $\epsilon$-corrupted set of $N$ frequency vectors is equivalent to that of robustly estimating the mean of a multinomial in $L_1$.

As such, it is natural to try to adapt existing algorithms for robust mean estimation of other distributions (in Euclidean norm); the fastest ones are based on the following simple filtering approach. Maintain weights for each point, initialized to uniform. At every step, measure the maximum "skew" of the weighted dataset in any direction, and if this skew is too high, update the weights by 1) finding the direction $v$ in which the corruptions skew the dataset the most, 2) giving a "score" to each point

based on how badly it skews the dataset in direction $v$, 3) downweighting or removing points with high scores. Otherwise, if the skew is low, output the empirical mean of the weighted dataset.

To prove correctness, one must show three things for the particular skewness measure and score function chosen: A) (*Regularity*) for any sufficiently large collection of $\epsilon$-corrupted batches, a particular deterministic regularity condition holds (Definition 3.2 and Lemma 3.3), B) (*Soundness*) under the regularity condition, if the skew of the weighted dataset is small, then the empirical mean of the weighted dataset is sufficiently close to the true mean (Lemma 3.5), C) (*Progress*) under the regularity condition, if the skew of the weighted dataset is large, then one iteration of the above update scheme will remove more weight from bad batches than from good (Lemma 3.6).

For isotropic Gaussians, skewness is just given by the maximum variance of the weighted dataset in any direction, i.e. $\max_{v\in\mathbb{S}^{n-1}}\langle vv^\top, \tilde{\Sigma}\rangle$ where $\tilde{\Sigma}$ is the empirical covariance of the weighted dataset. Given maximizing $v$, the "score" of a point $X$ is then simply its contribution to the skewness.

To learn in $L_1$ distance, the right set of test vectors $v$ to use is the Hamming cube $\{0,1\}^n$, so a natural attempt at adapting the above skewness measure to robust mean estimation of multinomials is to consider the quantity $\max_{v\in\{0,1\}^n}\langle vv^\top, \tilde{\Sigma}\rangle$. But one of the key challenges in passing from isotropic Gaussians to multinomial distributions is that this quantity is not very informative because we do not have a good handle on the covariance of $\mathrm{Mul}_k(\mu)$. In particular, it could be that for a direction $v$, $\langle vv^\top, \tilde{\Sigma}\rangle$ is high simply because the good points have high variance to begin with.

**An SDP for Skewness**  The clever workaround of [JO19] was to observe that we know exactly what the projection of a multinomial distribution $\mathrm{Mul}_k(\mu)$ in any $\{0,1\}^n$ direction $v$ is, namely $\mathrm{Bin}(k,\langle v,\mu\rangle)$. And so to discern whether the corrupted points skew our estimate in a given direction $v$, one should measure not the variance in the direction $v$, but rather the following *corrected quantity*: the variance in the direction $v$, *minus* what the variance would be if the distribution of the projections in the $v$ direction were actually given by $\mathrm{Bin}(k,\langle v,\tilde{\mu}\rangle)$, where $\tilde{\mu}$ is the empirical mean of the weighted dataset. We call this latter quantity a *variance proxy*. The new skewness measure can be written as

$$\max_{v\in\{0,1\}^n}\left\{\langle vv^\top, \tilde{\Sigma}\rangle - \frac{1}{k}(\langle v,\tilde{\mu}\rangle - \langle v,\tilde{\mu}\rangle^2)\right\}. \tag{1}$$

Finding the direction $v\in\{0,1\}^n$ which maximizes this corrected quantity is some Boolean quadratic programming problem which can be solved approximately by solving the natural SDP relaxation and rounding to a Boolean vector $v$ using the machinery of [AN04]. Using this approach, [JO19] obtained a polynomial-time algorithm for learning *general* discrete distributions from untrusted batches.

**Structured Distributions: Beyond Boolean Test Vectors**  Learning structured distributions in the classical sense is well-understood: if a distribution $\mu$ is close in total variation distance to being $s$-piecewise degree-$d$, then to estimate $\mu$ in total variation distance it is enough to approximate $\mu$ in a much weaker norm which we will denote by $\|\cdot\|_{\mathcal{A}_K}$, where $K$ is a parameter that depends on $s$ and $d$. We review the details for this in Section 1.3.

The key challenge that [CLM19] had to address to port these techniques to the untrusted batches setting was that unlike the Hamming cube or $\mathbb{S}^{n-1}$, it is unclear how to optimize over the set of test vectors dual to the $\mathcal{A}_K$ norm. Combinatorially, this set is easy to characterize: $\|\mu - \hat{\mu}\|_{\mathcal{A}_K}$ is small if and only if $\langle \mu - \hat{\mu}, v\rangle$ is small for all $v\in\mathcal{V}_{2K}^n\subset\{\pm1\}^n$, where $\mathcal{V}_{2K}^n$ is the set of all $v\in\{\pm1\}^n$ with at most $2K$ sign changes when read as a vector from left to right.

A key observation in [CLM19] was that vectors with few sign changes admit sparse representations in the *Haar wavelet basis*, so instead of working with $\mathcal{V}_{2K}^n$, one can simply work with a convex relaxation of this Haar-sparsity constraint. As such, if we let $\mathcal{K}\subseteq\mathbb{R}^{n\times n}$ denote the relaxation of the set of $\{vv^\top|v\in\mathcal{V}_{2K}^n\}$ to all matrices $\Sigma$ whose Haar transforms are "analytically sparse" in some appropriate, convex sense (see Section 2 for a formal definition), then as this set of test matrices contains the set of test matrices $vv^\top$ for $v\in\mathcal{V}_{2K}^n$, it is enough to learn $\mu$ in the norm associated to $\mathcal{K}$, which is strictly stronger than the $\mathcal{A}_K$ norm.

Our goal then is to produce $\hat{\mu}$ for which $\|\hat{\mu}-\mu\|_{\mathcal{K}}\triangleq\sup_{\Sigma\in\mathcal{K}}\langle\Sigma,(\hat{\mu}-\mu)^{\otimes2}\rangle^{1/2}$ is small. Even though $\|\cdot\|_{\mathcal{K}}$ is a stronger norm, it turns out $\mathcal{K}$'s metric entropy is still quite small. As we elaborate on in the supplementary material, the analysis of this in [CLM19] left much room for tightening. A refined analysis of $\mathcal{K}$ in the present work allows us to get nearly tight sample complexity bounds.

**Putting Everything Together** All the pieces are in place to instantiate the filtering framework: in lieu of the quantity in (1), which can be phrased as the maximization of some quadratic $\langle vv^\top, M(w)\rangle$ over $\{\pm 1\}^n$, where $M(w) \in \mathbb{R}^{n \times n}$ depends on the dataset and the weights $w$ on its points,[1] we can define our skewness measure as $\max_{\Sigma \in \mathcal{K}} \langle \Sigma, M(w)\rangle = \|M(w)\|_{\mathcal{K}}$, and we can define the score for each point in the dataset to be its contribution to the skewness measure (see (3)).

The reader may be wondering why, unlike [JO19] or applications of filtering in other contexts, we never round $\Sigma$ to an actual vector $v \in \mathcal{V}_{2K}^n$ before computing skewness and scores. We emphasize that once restricted to bounded sign change vectors $v \in \mathcal{V}_{2K}^n$, the optimization problem in (1) becomes significantly harder. Indeed, optimizing general quadratic forms $v^\top A v$ over $\mathcal{V}_{2K}^n$ can essentially capture the problem of densest $2K$-subgraph: for any $A'$, take $A \triangleq T^\top A' T$ where $T$ is the matrix which maps $v$ to $(0, v_2 - v_1, v_3 - v_2, ..., v_n - v_{n-1})$. Then optimizing $v^\top A v$ over $v \in \mathcal{V}_{2K}^n$ is equivalent to optimizing $w^\top A' w$ over $\ell$-sparse vectors $w \in \{0, \pm 1\}^n$ whose nonzero entries are alternating in sign. Modulo this alternating sign condition, this is at least as hard as densest $\ell$-subgraph, for which it is impossible, under the Exponential Time Hypothesis, to efficiently achieve any sub-poly$(n)$ factor approximation [Man17]. In contrast, for the original optimization problem (1) considered in [JO20], one can reduce to the question of computing cut-norm, which can be done to within a constant factor via Krivine rounding.[AN04]

As our subsequent analysis will show, it turns out that rounding is unnecessary, both in our setting and in the unstructured distribution setting of [JO19]. This should be quite surprising as the afore-mentioned hardness of approximation [Man17] suggests that the integrality gap of our relaxation should be terrible. In particular, the norm $\|\cdot\|_{\mathcal{K}}$ induced by our relaxation might be very distorted relative to the $\mathcal{A}_\ell$ norm we actually care about. However, if one examines the proof ingredients enumerated above, it becomes evident that the filtering framework does not actually require finding a concrete direction in $\mathbb{R}^n$ in which to filter, merely a skewness measure and score functions under which regularity, soundness, and progress can be proven. That said, it becomes more challenging to prove these ingredients when $\Sigma$ is not rounded to an actual direction, though nevertheless possible.

We hope that this observation will prove useful in future applications of filtering.

## 1.2 Related Work

The problem of learning from untrusted batches was motivated by problems in reliable distributed learning and federated learning [MMR+17, KMY+16]. The general question of learning from batches has been considered in a number of settings [LRR13, TKV17] in theory, but these algorithms do not work in the presence of adversarial noise.

The study of univariate shape constrained density estimation has a long history in statistics and computer science, and we cannot hope to do justice to it here. See [BBBB72] for a survey of classical results in the area, and [O'B16, Dia16] for a survey of more recent results in this area. Of particular relevance to us are the techniques based on the classical piecewise polynomial (or spline) methods, see e.g. [WW83, Sto94, SHKT97, WN07]. Recent work, which we build off of, demonstrates that this framework is capable of achieving nearly-optimal sample complexity and runtime, for a large class of structured distributions [CDSS13, CDSS14b, CDSS14a, ADH+15, ADLS17].

Our techniques are also related to a recent line of work on robust statistics [DKK+19, LRV16, CSV17, DKK+17, HL18, KSS18], a classical problem dating back to the 60s and 70s [Ans60, Tuk60, Hub92, Tuk75]. See [Li18, Ste18, DK19] for a more comprehensive survey of this line of work.

Concurrently and independently of this work, a newer work of Jain and Orlitsky [JO20] obtains very similar results, though our quantitative guarantees are incomparable: the number of batches $N$ they need scales linearly in $s \cdot d$ and independently of $n$, but also scales with $\sqrt{k}$ and $1/\epsilon^3$.

## 1.3 Technical Preliminaries

**Notation** Let $\Delta^n \subset \mathbb{R}^n$ be the simplex of probability distributions over $[n]$. Given $\mu \in \Delta^n$, let $\mathrm{Mul}_k(\mu)$ denote the distribution over $\Delta^n$ given by sampling a frequency vector from the multinomial distribution arising from $k$ draws from the distribution over $[n]$ specified by $\mu$, and dividing by $k$.

Given matrix $M \in \mathbb{R}^{n \times n}$, let $\|M\|_{\max}$ denote the maximum absolute value of any entry in $M$, let $\|M\|_{1,1}$ denote the absolute sum of its entries, and let $\|M\|_F$ denote its Frobenius norm.

Given batches $X_1, \cdots, X_N \sim \mathrm{Mul}_k(\mu)$ and $U \subseteq [N]$, define $w(U) : [N] \to [0, 1/N]$ to be the set of weights which assigns $1/N$ to all points in $U$ and $0$ to all other points. Also define its normalization $\hat{w}(U) \triangleq w(U)/\|w\|_1$. Let $\mathcal{W}_\epsilon$ denote the set of weights $w : [N] \to [0, 1/N]$ which are convex combinations of such weights for $|U| \geq (1 - \epsilon)N$. Given $w$, define $\mu(w) \triangleq \sum_{i=1}^N \frac{w_i}{\|w\|_1} X_i$, and define $\mu(U) \triangleq \mu(w(U))$, that is, the empirical mean of the batches indexed by $U$.

Given batches $X_1, \cdots, X_N \sim \mathrm{Mul}_k(\mu)$, weights $w$, and $\nu_1, ..., \nu_N \in \Delta^n$, define the matrices

$$A(w, \{\nu_i\}) = \sum_{i=1}^N w_i (X_i - \nu_i)^{\otimes 2} \quad \text{and} \quad B(\{\nu_i\}) = \frac{1}{N} \sum_{i=1}^N \mathbb{E}_{X \sim \mathrm{Mul}_k(\nu_i)} [(X - \nu_i)^{\otimes 2}].$$

When $\nu_1 = \cdots = \nu_N = \nu$, denote these matrices by $A(w, \nu)$ and $B(\nu)$ and note that

$$B(\nu) = \frac{1}{k} \left( \mathrm{diag}(\nu) - \nu^{\otimes 2} \right)$$

Also define $M(w, \{\nu_i\})) \triangleq A(w, \{\nu_i\}) - B(\{\nu_i\})$ and $M(w, \nu) \triangleq A(w, \nu) - B(\nu)$. We will also denote $M(w, \mu(w))$ by $M(w)$ and $M(\hat{w}(U))$ by $M_U$.

**Generative Model** Let $\epsilon, \omega > 0$, $n, k, N \in \mathbb{N}$, and let $\mu \in \Delta^n$ be any distribution over $[n]$. $Y_1, ..., Y_N$ is an $\epsilon$-*corrupted* $\omega$-*diverse* set of $N$ batches of size $k$ from $\mu$ if it is generated as follows:

1. For every $i \in [(1 - \epsilon)N]$, $\tilde{Y}_i = (\tilde{Y}_i^1, ..., \tilde{Y}_i^k)$ is a set of $k$ iid draws from $\mu_i$, where $\mu_i \in \Delta^n$ is some probability distribution over $[n]$ for which $d_{\mathrm{TV}}(\mu, \mu_i) \leq \omega$.

2. A computationally unbounded adversary inspects $\tilde{Y}_1, ..., \tilde{Y}_{(1-\epsilon)N}$ and adds $\epsilon N$ arbitrarily chosen $\tilde{Y}_{(1-\epsilon)N+1}, ..., \tilde{Y}_N \in [n]^k$, and returns the entire collection of tuples in any order as $Y_1, ..., Y_N$.

Let $S_G, S_B \subset [N]$ denote the indices of the uncorrupted (good) and corrupted (bad) batches. Given batch $Y_i \in [n]^k$, define $X_i \in \Delta^n$ to be the frequency vector whose $a$-th entry is $\frac{1}{k} \sum_{j=1}^k \mathbb{1}[Y_i^j = a]$. For each $i \in S_G$, $X_i$ is i.i.d. from $\mathrm{Mul}_k(\mu_i)$. We work solely in this frequency vector perspective.

$\mathcal{A}_K$ **Norms and VC Complexity** We review basics about learning structured distributions.

**Definition 1.2** ($\mathcal{A}_K$ norms, see e.g. [DL01]). *For positive integers $K \leq n$, define $\mathcal{A}_K$ to be the set of all unions of at most $K$ disjoint intervals over $[n]$, where an interval is any subset of $[n]$ of the form $\{a, a + 1, \cdots, b - 1, b\}$. The $\mathcal{A}_K$ distance between two distributions $\mu, \nu$ over $[n]$ is $\|\mu - \nu\|_{\mathcal{A}_K} = \max_{S \in \mathcal{A}_K} |\mu(S) - \mu(S)|$. Equivalently, say that $v \in \{\pm 1\}^n$ has $2K$ sign changes if there are exactly $2K$ indices $i \in [n - 1]$ for which $v_{i+1} \neq v_i$. Then if $\mathcal{V}_{2K}^n$ denotes the set of all such $v$, we have $\|\mu - \nu\|_{\mathcal{A}_K} = \frac{1}{2} \max_{v \in \mathcal{V}_{2K}^n} \langle \mu - \nu, v \rangle$. Note that $\|\cdot\|_{\mathcal{A}_1} \leq \|\cdot\|_{\mathcal{A}_2} \leq \cdots \leq \|\cdot\|_{\mathcal{A}_{n/2}} = \|\cdot\|_{TV}$.*

**Definition 1.3.** *A distribution $\mu$ over $[n]$ is $(\eta, s)$-piecewise degree-$d$ if there is a partition of $[n]$ into $t$ disjoint intervals $\{[a_i, b_i]\}_{1 \leq i \leq t}$, together with univariate degree-$d$ polynomials $r_1, \cdots, r_t$ and distribution $\mu'$, such that $d_{TV}(\mu, \mu') \leq \eta$ and, for all $i \in [t]$, $\mu'(x) = r_i(x)$ for all $x \in [n]$ in $[a_i, b_i]$.*

A proof of the following lemma, a consequence of [ADLS17], can be found in [CLM19].

**Lemma 1.4** (Lemma 5.1 in [CLM19], follows by [ADLS17]). *Let $K = s(d + 1)$. If $\mu$ is $(\eta, s)$-piecewise degree-$d$ and $\|\mu - \hat{\mu}\|_{\mathcal{A}_K} \leq \zeta$, then there is an algorithm which, given the vector $\hat{\mu}$, outputs a distribution $\mu^*$ for which $d_{TV}(\mu, \mu^*) \leq 2\zeta + 4\eta$ in time $\mathrm{poly}(s, d, 1/\eta)$.*

In light of this, to show Theorem 1.1 it suffices to get good error in $\mathcal{A}_{\ell/2}$ norm, where

$$\ell \triangleq 2s(d + 1). \tag{2}$$

## 2 Semidefinite Program for Finding the Direction of Largest Variance

**Haar Wavelets and $\mathcal{V}_\ell^n$** We briefly recall the definition of Haar wavelets. See [CLM19] for details.

**Definition 2.1.** *Let $m$ be a positive integer and let $n = 2^m$. The* Haar wavelet basis *is an orthonormal basis over $\mathbb{R}^n$ consisting of the* father wavelet $\psi_{0_{father},0} = n^{-1/2} \cdot \mathbf{1}$, *the* mother wavelet $\psi_{0_{mother},0} = n^{-1/2} \cdot (1, \cdots, 1, -1, \cdots, -1)$ *(where $(1, \cdots, 1, -1, \cdots, -1)$ contains $n/2$ 1's and $n/2$ -1's), and for every $i, j$ for which $1 \le i < m$ and $0 \le j < 2^i$, the wavelet $\psi_{i,j}$ whose $2^{m-i} \cdot j + 1, \cdots, 2^{m-i} \cdot j + 2^{m-i-1}$-th coordinates are $2^{-(m-i)/2}$ and whose $2^{m-i} \cdot j + (2^{m-i-1}+1), \cdots, 2^{m-i} \cdot j + 2^{m-i}$-th coordinates are $-2^{-(m-i)/2}$, and whose remaining coordinates are 0.*

We will use the following notation when referring to Haar wavelets:

- Let $H_m$ denote the $n \times n$ matrix whose rows consist of the vectors of the Haar wavelet basis for $\mathbb{R}^n$. When the context is clear, we will omit the subscript and refer to this matrix as $H$.

- For $\nu \in [n]$, if the $\nu$-th Haar wavelet in $\mathbb{R}^n$ is $\psi_{i,j}$, define the weight $\mathbf{h}^{(\nu)} \triangleq 2^{-(m-i)/2}$.

- For any index $i \in \{0_{father}, 0_{mother}, 1, \cdots, m-1\}$, let $T_i \subset [n]$ denote the set of indices $\nu$ for which the $\nu$-th Haar wavelet is of the form $\psi_{i,j}$ for some $j$.

- Given any $p \ge 1$, define the *Haar-weighted $L^p$ norm* $\|\cdot\|_{p;\mathbf{h}}$ on $\mathbb{R}^n$ by $\|w\|_{p;\mathbf{h}} \triangleq \|w'\|_p$, where for every $a \in [n]$, $w'_a \triangleq \mathbf{h}^{(a)} w_a$. Likewise, given any norm $\|\cdot\|_*$ on $\mathbb{R}^{n \times n}$, define the Haar-weighted $*$-norm $\|\cdot\|_{*;\mathbf{h}}$ on $\mathbb{R}^{n \times n}$ by $\|\mathbf{M}\|_{*;\mathbf{h}} \triangleq \|\mathbf{M}'\|_*$, where for every $a, b \in [n]$, $\mathbf{M}'_{a,b} \triangleq \mathbf{h}^{(a)} \mathbf{h}^{(b)} \mathbf{M}_{a,b}$. We refer to Section 2.1 of [CLM19] for a discussion of why this weighting is crucial.

The key observation is that any $v \in \{\pm 1\}^n$ with at most $\ell$ sign changes, where $\ell$ is given by (2), has an $(\ell \log n + 1)$-sparse representation in the Haar wavelet basis. We will use the following fundamental fact about Haar wavelets, a cruder version of which appears as Lemma 6.3 in [CLM19].

**Lemma 2.2.** *Let $v \in \mathcal{V}_\ell^n$. Then $Hv$ has at most $\ell \log n + 1$ nonzero entries, and furthermore $\|Hv\|_{\infty;\mathbf{h}} \le 1$. In particular, $\|Hv\|_{2;\mathbf{h}}^2, \|Hv\|_{1;\mathbf{h}} \le \ell \log n + 1$.*

**Relaxing to Analytic Sparsity**  Recall that in [JO19], the authors consider the binary optimization problem $\max_{v \in \{0,1\}^n} |v^\top M_U v|$. We would like to form a convex relaxation of the more challenging optimization problem $\max_{v \in \mathcal{V}_\ell^n} |v^\top M_U v|$. Motivated by Lemma 2.2, we consider the following:

**Definition 2.3.** *Let $\ell$ be given by (2). Let $\mathcal{K}$ denote the (convex) set of matrices $\Sigma \in \mathbb{R}^{n \times n}$ for which 1) $\Sigma \succeq 0$, 2) $\|\Sigma\|_{\max} \le 1$, 3) $\|H\Sigma H^\top\|_{1,1;\mathbf{h}} \le \ell \log n + 1$, 4) $\|H\Sigma H^\top\|_{F;\mathbf{h}}^2 \le \ell \log n + 1$, and 5) $\|H\Sigma H^\top\|_{\max;\mathbf{h}} \le 1$. Let $\|\cdot\|_{\mathcal{K}}$ denote the associated norm given by $\|\mathbf{M}\|_{\mathcal{K}} \triangleq \sup_{\Sigma \in \mathcal{K}} |\langle \mathbf{M}, \Sigma \rangle|$. By abuse of notation, for vectors $v \in \mathbb{R}^n$ we will also use $\|v\|_{\mathcal{K}}$ to denote $\|vv^\top\|_{\mathcal{K}}^{1/2}$.*

*Because $\mathcal{K}$ has an efficient separation oracle, one can compute $\|\cdot\|_{\mathcal{K}}$ in polynomial time.*

A standard way to relax a sparsity constraint of the form $\|v\|_0 \le s$ is to require "analytic sparsity," namely that $\|v\|_1^2 / \|v\|_2^2 \le s$. Constraints 3 and 4 should be thought of as a matrix analogue of analytic sparsity in the Haar basis. Constraint 2 is to ensure that $\|\cdot\|_{\mathcal{K}}$ is weaker than $L_1$. Constraint 5 will appear in a delicate way in the proof of the metric entropy bound (Lemma 3.4).

**Remark 2.4.** *Besides not being a sum-of-squares program like the one considered in [CLM19], this relaxation is also different because of Constraints 3 and 5. These will be crucial for getting refined sample complexity bounds (see the proof of Lemma 3.4 in the supplementary material).*

**Corollary 2.5** (Corollary of Lemma 2.2). *$vv^\top \in \mathcal{K}$ for any $v \in \mathcal{V}_\ell^n$.*

## 3   Filtering Algorithm and Analysis

In this section we describe our algorithm LEARNWITHFILTER and prove Theorem 1.1. We will maintain weights $w : [N] \to \mathbb{R}_{\ge 0}$ for each of the batches. In every iteration, we compute $\Sigma \in \mathcal{K}$ maximizing $|\langle \bar{M}(w), \Sigma \rangle|$. If $|\langle \bar{M}(w), \Sigma \rangle| \le O\left(\frac{\epsilon}{k} \log 1/\epsilon\right)$, then output $\mu(w)$. Otherwise, update the weights as follows: for every batch $X_i$, compute the score $\tau_i$ given by

$$\tau_i \triangleq \left\langle (X_i - \mu(w))^{\otimes 2}, \Sigma \right\rangle, \tag{3}$$

and set the weights to be the output of a basic univariate filtering subroutine, 1DFILTER($\tau, w$). The pseudocode for LEARNWITHFILTER and 1DFILTER is given in Algorithm 1 and 2.

---

**Algorithm 1:** LEARNWITHFILTER(batches $\{X_i\}_{i\in[N]}$, corruption fraction $\epsilon$)

---

1 Initialize $w \leftarrow w([N])$
2 **while** $\|M(w)\|_{\mathcal{K}} \geq \Omega(\omega + \frac{\epsilon}{k}\log 1/\epsilon)$ **do**
3     $\Sigma \leftarrow \operatorname{argmax}_{\Sigma' \in \mathcal{K}} |\langle M(w), \Sigma\rangle|$
4     Compute scores $\tau : [N] \to \mathbb{R}_{\geq 0}$ according to (3).
5     $w \leftarrow$ 1DFILTER($\tau, w$)
6 Using the algorithm of [ADLS17] (Lemma 1.4), **return** the $s$-piecewise, degree-$d$ distribution $\hat{w}$ minimizing $\|\mu(w) - \hat{\mu}\|_{s(d+1)}$ (up to additive error $\eta$).

---

The basic guarantee of 1DFILTER is that if the weighted average of the scores of the good points is less than that of the bad, then it will decrement the weights so that the bad points lose more weight than the good points overall:[2]

**Lemma 3.1.** *For scores $\tau : [N] \to \mathbb{R}_{\geq 0}$ and weights $w : [N] \to \mathbb{R}_{\geq 0}$, if $\sum_{i\in S_G} w_i\tau_i < \sum_{i\in S_B} w_i\tau_i$, then the output $w'$ of 1DFILTER($\tau, w$) satisfies (a) $w'_i \leq w_i$ for all $i \in [N]$, (b) the support of $w'$ is a strict subset of the support of $w$, and (c) $\sum_{i\in S_G} w_i - w'_i < \sum_{i\in S_B} w_i - w'_i$.*

---

**Algorithm 2:** 1DFILTER(scores $\tau$, weights $w$)

---

1 **return** weights $w'$ given by $w'_i \leftarrow \left(1 - \frac{\tau_i}{\tau_{\max}}\right)w_i$ for all $i \in [N]$, where $\tau_{\max} \leftarrow \max_{i:w_i>0} \tau_i$

---

We now formally state what we need in terms of the three main ingredients (regularity, soundness, progress) sketched in Section 1.1.

**Regularity**    We need the following deterministic conditions to hold:

**Definition 3.2** ($\epsilon$-goodness)**.** *Take a set of points $U \subset [N]$, and let $\{\mu_i\}_{i\in U}$ be a collection of distributions over $[n]$. For any $W \subseteq U$, define $\overline{\mu}_W \triangleq \frac{1}{|W|}\sum_{i\in W} \mu_i$. Denote $\overline{\mu} \triangleq \overline{\mu}_U$.*

*We say $U$ is $\epsilon$-good if it satisfies that for all $W \subset U$ for which $|W| = \epsilon|U|$,*

1. *(Conc. of mean) $\|\mu(U) - \overline{\mu}\|_{\mathcal{K}} \leq O(\frac{\epsilon}{\sqrt{k}}\sqrt{\log 1/\epsilon})$, $\|\mu(W) - \overline{\mu}_W\|_{\mathcal{K}} \leq O(\frac{1}{\sqrt{k}}\sqrt{\log 1/\epsilon})$*

2. *(Conc. of covar.) $\|M(\hat{w}(U), \{\mu_i\}_{i\in U})\|_{\mathcal{K}} \leq O(\frac{\epsilon \log 1/\epsilon}{k})$, $\|A(\hat{w}(W), \{\mu_i\}_{i\in W}\|_{\mathcal{K}} \leq O(\frac{\log 1/\epsilon}{k})$*

3. *(Conc. of variance proxy) $\|B(\hat{\mu}(U)) - B(\{\mu_i\}_{i\in U})\|_{\mathcal{K}} \leq O(\omega^2/k + \epsilon/k)$*

*Note, we ignore the variance proxy for $W$ in 2. because its contribution is negligible, as $|W| = \epsilon|U|$. Also, when $\omega > 0$, we need another condition which is rather technically involved, see the supplement.*

**Lemma 3.3** (Regularity)**.** *If $U$ is a set of $\widetilde{\Omega}\left(\log(1/\delta)(\ell^2/\epsilon^2)\cdot\log^3(n)\right)$ independent batches from $Mul_k(\mu_1), ..., Mul_k(\mu_{|U|})$, then $U$ is $\epsilon$-good with probability at least $1 - \delta$.*

The key component in the proof of Lemma 3.3 that lets us get sample complexity quadratic in $\ell$ is the existence of a suitable net over $\mathcal{K}$.

**Lemma 3.4.** *For every $0 < \eta \leq 1$, there exists a net $\mathcal{N} \subset \mathbb{R}^{n\times n}$ of size $O(n^3\ell^2\log^2 n/\eta)^{(\ell\log n+1)^2}$ of matrices such that for every $\Sigma \in \mathcal{K}$, there exists some $\tilde{\Sigma} = \sum_\nu \Sigma^*_\nu$ for $\Sigma^*_\nu \in \mathcal{N}$ such that the following holds: 1) $\|\Sigma - \tilde{\Sigma}\|_F \leq \eta$, 2) $\sum_\nu \alpha_\nu \leq 1$, and 3) $\|\Sigma^*_\nu\|_{\max} \leq O(1)$.*

**Soundness**    For soundness, we will show the following key geometric property, namely a bound on the accuracy of an estimate $\mu(w)$ given by weights $w$ in terms of $\|M(w)\|_{\mathcal{K}}$. It can be interpreted as saying that if the skewness $\|M(w)\|_{\mathcal{K}}$ of the dataset weighted by $w$ is low, then the weighted average $\mu(w)$ is close to the empirical mean of the good points. The contrapositive of this says that if the weighted average is still far from the empirical mean of the good points, then the solution to the SDP in the definition of $\|\cdot\|_{\mathcal{K}}$ is a certificate that this is the case.

**Lemma 3.5** (Soundness). *If $S_G$ is $\epsilon$-good and $|S_G| \geq (1 - \epsilon)N$, then for any $w \in \mathcal{W}_\epsilon$,*

$$\left\| \mu(w) - \frac{1}{|S_G|} \sum_{i \in S_G} \mu_i \right\|_\mathcal{K} \leq O\left( \frac{\epsilon}{\sqrt{k}} \sqrt{\log 1/\epsilon} + \epsilon \cdot \omega + \sqrt{\epsilon \left( \|M(w)\|_\mathcal{K} + \omega^2 + \frac{\epsilon}{k} \log 1/\epsilon \right)} \right).$$

We emphasize that because we are working directly with the SDP-based norm $\| \cdot \|_\mathcal{K}$ and avoiding any sort of rounding, the proof of Lemma 3.5 introduces various technical complications that do not manifest in analogous results for other settings of robust mean estimation [Li18, DKK+19].

**Progress**   The following says as long as we remain in the main loop of LEARNWITHFILTER and have so far thrown out more bad weight than good, we will continue to do so in the next iteration.

**Lemma 3.6** (Progress). *Let $w$ and $w'$ be the weights at the start and end of a single iteration of the main loop of LEARNWITHFILTER. There is an absolute constant $C > 0$ such that if $\|M(w)\|_\mathcal{K} > C \cdot \frac{\epsilon}{k} \log 1/\epsilon$ and $\sum_{i \in S_G} \frac{1}{N} - w_i < \sum_{i \in S_B} \frac{1}{N} - w_i$, then $\sum_{i \in S_G} w_i - w'_i < \sum_{i \in S_B} w_i - w'_i$.*

We can now combine Lemma 3.5 and Lemma 3.6 to get a proof of Theorem 1.1.

*Proof of Theorem 1.1.* Let $\hat{\mu}$ be the output of LEARNWITHFILTER. By Lemma 1.4, the definition of $\| \cdot \|_{\ell/2}$, and Corollary 2.5, it is enough to show that $\|\hat{\mu} - \mu\|_\mathcal{K} \leq O(\omega + \frac{\epsilon}{\sqrt{k}} \sqrt{\log 1/\epsilon})$. By Lemma 3.5 and the termination condition of the loop in LEARNWITHFILTER, we just need to show that the algorithm terminates (in polynomial time) and that $w \in \mathcal{W}_{O(\epsilon)}$. But by induction and Lemma 3.6, every iteration removes more bad weight than good, and by Lemma 3.1, the support of $w$ goes down by at least one every time 1DFILTER is run. So the loops terminates after at most $N$ iterations, each of which can be implemented in polynomial time. At the end, at most $\epsilon$ fraction of the total mass on $S_G$ has been removed, so the final weights $w$ satisfy $w \in \mathcal{W}_{2\epsilon}$. $\qquad\square$

## 4   Numerical Experiments

We empirically evaluated our algorithm on synthetic data. We compared our algorithm LEARN-WITHFILTER, the empirical mean of all samples, and the "oracle" i.e. the empirical mean of the uncorrupted samples (in Figures 1a and 1b, these are labeled "filter", "naive", "oracle" respectively).

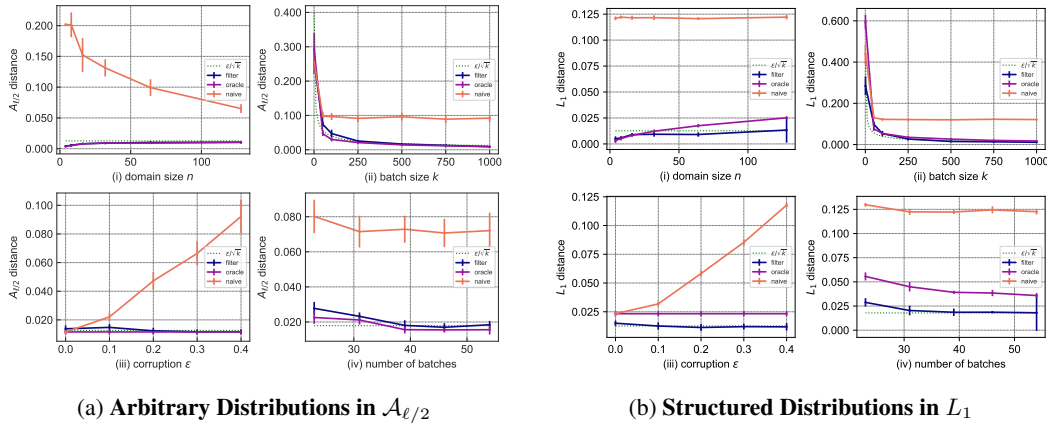

(a) **Arbitrary Distributions in $\mathcal{A}_{\ell/2}$**    (b) **Structured Distributions in $L_1$**

Figure 1: Synthetic experiments

By definition, the oracle dominates the algorithms of [CLM19, JO19] for the *unstructured* case, as they search for a subset of the data and output that subset's empirical mean. But by Theorem 1.1, LEARNWITHFILTER should *outperform* the oracle when $\mu$ is *structured* and there are too few samples for the empirical mean of the good batches to concentrate. We confirm this empirically.

Our experiments fall under two types: $(a)$ learning an *arbitrary distribution* in $\mathcal{A}_\ell$ norm and B) learning a *structured distribution* in *total variation distance*. Experiments of type $(a)$ (resp. $(b)$) show that LEARNWITHFILTER can be used to learn from untrusted batches in $\mathcal{A}_\ell$ norm even for distributions which are not structured (resp. that LEARNWITHFILTER can outperform the oracle).

Throughout, $\omega = 0$ and $\ell = 10$. For the domain sizes $n$ we work with, enumerating over $\mathcal{V}_\ell^n$ would be prohibitively expensive, necessitating the use of our SDP.

For each trial, we randomly generated $\mu$, either by sampling uniformly from $[0,1]^n$ for $(a)$ or by choosing a random $\ell$-wise constant function, and normalizing. We drew the corrupted batches from $\text{Mul}_k(\nu)$ for $\nu$ chosen at an appropriate distance from $\mu$. We examined the effect of varying one of the following four parameters at a time: (i) domain size $n$, (ii) batch size $k$, (iii) corruption fraction $\epsilon$, and (iv) total number of batches $N$. Each data point in Figure 1 corresponds to a median of ten trials.

In (i), we fixed $k = 1000$, $\epsilon = 0.4$, $N \approx \ell/\epsilon^2$. Note that while $N$ is independent of $n$, our algorithm is competitive with the oracle for type $(a)$ and superior for type $(b)$. In (ii), we fixed $\epsilon = 0.4$, $n = 64$, $N \approx \ell/\epsilon^2$. Note that while our algorithm's error and the oracle's error decay with $k$, the empirical mean's error remains fixed. In (iii) we fixed $n = 64$, $k = 1000$, $N \approx \ell/0.4^2$; in (iv) we fixed $n = 128$, $k = 1000$, $\epsilon = 0.4$. Again, we are either competitive with or superior to the oracle.

The experiments were conducted on a MacBook Pro with 2.6 GHz Dual-Core Intel Core i5 processor and 8 GB of RAM. For the implementation, we used the SCS solver in CVXPY for our semidefinite programs. Over a domain of size 128, LEARNWITHFILTER takes roughly 7-10 minutes. See the supplement for further implementation details. All code can be found at `https://github.com/secanth/federated`.

## Broader Impact

The goal for this work is to lay theoretical foundations for some basic problems in federated learning. As such, it may be of general societal benefit because it may lead to better systems for pooling data that cannot be manipulated by small groups with ulterior motives. Our algorithms do not leverage biases in data, but on the contrary seek to efficiently identify them and mitigate their effect. The main negative is that even algorithms with provable guarantees can be used outside of settings they are intended, in which case they can have unpredictable behavior. However our theoretical analysis also provides guidance on when using our algorithms ought to be appropriate.

## Acknowledgments

The authors would like to thank Ayush Jain and Alon Orlitsky for agreeing to coordinate the initial online submission of their preprint [JO20] with ours. S.C. was supported by a Paul and Daisy Soros Fellowship, NSF CAREER Award CCF-1453261, and NSF Large CCF-1565235. A.M. was supported in part by a Microsoft Trustworthy AI Grant, NSF CAREER Award CCF-1453261, NSF Large CCF-1565235, a David and Lucile Packard Fellowship, an Alfred P. Sloan Fellowship and an ONR Young Investigator Award.

## Footnotes

[1]Note that we have switched to $\{\pm 1\}^n$ in place of $\{0, 1\}^n$. The difference turns out to be immaterial, so we do not belabor this point, and the former is more convenient for understanding how we handle $\mathcal{V}_{2K}^n \subset \{\pm 1\}^n$.

[2]See Lemma 4.5 from [CSV17] or Lemma 17 from [SCV18] for similar downweighting schemes and analysis.

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
