[Supplementary Material 1 · supplement.pdf]

# Supplementary Materials Roadmap

In Section 1, we overview notation, formally define our generative model, give miscellaneous technical tools, and review the basics on classical learning of structured distributions and on Haar wavelets. In Section 2, we define the semidefinite program that we use to compute skewness. In Section 3, we give our algorithm LEARNWITHFILTER and prove our main result, Theorem 1.1, restated here for convenience:

**Theorem 1.1.** *Let $\mu$ be a distribution over $[n]$ which is $\eta$-approximated by an $s$-part piecewise polynomial with degree at most $d$. There is an algorithm which runs in time polynomial in all parameters and estimates $\mu$ to within $O\left(\eta + \omega + \frac{\epsilon}{\sqrt{k}}\sqrt{\log 1/\epsilon}\right)$ in total variation after drawing $N$ $\epsilon$-corrupted batches, each of size $k$, where $N = \widetilde{O}\left((s^2 d^2/\epsilon^2) \cdot \log^3(n)\right)$ is the number of batches.*

In Section 4, we describe our empirical evaluations of LEARNWITHFILTER on synthetic data. In Appendices A, B, and C, we complete the proofs of some deferred technical statements relating to deterministic regularity conditions and metric entropy bounds.

# 1 Technical Preliminaries

## 1.1 Notation

- Given $p \in [0,1]$, let $\text{Bin}(k,p)$ denote the *normalized* binomial distribution, which takes values in $\{0, 1/k, \cdots, 1\}$ rather than $\{0, 1, \cdots, k\}$.

- Let $\Delta^n \subset \mathbb{R}^n$ be the simplex of nonnegative vectors whose coordinates sum to 1. Any $p \in \Delta^n$ naturally corresponds to a probability distribution over $[n]$.

- Let $\mathbf{1}_n \in \mathbb{R}^n$ denote the all-ones vector. We omit the subscript when the context is clear.

- Given matrix $M \in \mathbb{R}^{n \times n}$, let $\|M\|_{\max}$ denote the maximum absolute value of any entry in $M$, let $\|M\|_{1,1}$ denote the absolute sum of its entries, and let $\|M\|_F$ denote its Frobenius norm.

- Given $\mu \in \Delta^n$, let $\text{Mul}_k(\mu)$ denote the distribution over $\Delta^n$ given by sampling a frequency vector from the multinomial distribution arising from $k$ draws from the distribution over $[n]$ specified by $\mu$, and dividing by $k$.

- Given samples $X_1, \cdots, X_N \sim \text{Mul}_k(\mu)$ and $U \subseteq [N]$, define $w(U) : [N] \to [0, 1/N]$ to be the set of weights which assigns $1/N$ to all points in $U$ and 0 to all other points. Also define its normalization $\hat{w}(U) \triangleq w(U)/\|w\|_1$. Let $\mathcal{W}_\epsilon$ denote the set of weights $w : [N] \to [0, 1/N]$ which are convex combinations of such weights for $|U| \geq (1-\epsilon)N$. Given $w$, define $\mu(w) \triangleq \sum_{i=1}^N \frac{w_i}{\|w\|_1} X_i$, and define $\mu(U) \triangleq \mu(w(U))$, that is, the empirical mean of the samples indexed by $U$.

- Given samples $X_1, \cdots, X_N \sim \text{Mul}_k(\mu)$, weights $w$, and $\nu_1, ..., \nu_N \in \Delta^n$, define the matrices

$$A(w, \{\nu_i\}) = \sum_{i=1}^N w_i (X_i - \nu_i)^{\otimes 2} \quad \text{and} \quad B(\{\nu_i\}) = \frac{1}{N} \sum_{i=1}^N \mathbb{E}_{X \sim \text{Mul}_k(\nu_i)} [(X - \nu_i)^{\otimes 2}].$$

When $\nu_1 = \cdots = \nu_N = \nu$, denote these matrices by $A(w, \nu)$ and $B(\nu)$ and note that

$$B(\nu) = \frac{1}{k} \left(\text{diag}(\nu) - \nu^{\otimes 2}\right). \tag{1}$$

Also define $M(w, \{\nu_i\})) \triangleq A(w, \{\nu_i\}) - B(\{\nu_i\})$ and $M(w, \nu) \triangleq A(w, \nu) - B(\nu)$. We will also denote $M(w, \mu(w))$ by $M(w)$ and $M(\hat{w}(U))$ by $M_U$.

To get intuition for these definitions, note that any bitstring $v \in \{0,1\}^n$ corresponding to $S \subseteq [n]$ induces a normalized binomial distribution $Y \triangleq \text{Bin}(n, \langle \mu, v \rangle) \in [0,1]$, and any sample $X_i \sim \text{Mul}_k(\mu)$ induces a corresponding sample $\langle X_i, v \rangle$ from $Y$. Then $\langle vv^\top, M_U \rangle$ is the difference between the empirical variance of $Y$ and the variance of the binomial distribution $\text{Bin}(n, \langle \mu(U), v \rangle)$.

 ## 1.2 The Generative Model

41 Throughout the rest of the paper, let $\epsilon, \omega > 0$, $n, k, N \in \mathcal{N}$, and let $\mu$ be some probability distribution
42 over $[n]$.

43 **Definition 1.1.** *We say $Y_1, ..., Y_N$ is an $\epsilon$-corrupted $\omega$-diverse set of $N$ batches of size $k$ from $\mu$ if*
44 *they are generated via the following process:*

45 • *For every $i \in [(1-\epsilon)N]$, $\tilde{Y}_i = (\tilde{Y}_i^1, ..., \tilde{Y}_i^k)$ is a set of $k$ iid draws from $\mu_i$, where $\mu_i \in \Delta^n$*
46 *is some probability distribution over $[n]$ for which $d_{TV}(\mu, \mu_i) \leq \omega$.*

47 • *A computationally unbounded adversary inspects $\tilde{Y}_1, ..., \tilde{Y}_{(1-\epsilon)N}$ and adds $\epsilon N$ arbitrarily*
48 *chosen tuples $\tilde{Y}_{(1-\epsilon)N+1}, ..., \tilde{Y}_N \in [n]^k$, and returns the entire collection of tuples in any*
49 *arbitrary order as $Y_1, ..., Y_N$.*

50 *Let $S_G, S_B \subset [N]$ denote the indices of the uncorrupted (good) and corrupted (bad) batches.*

51 It turns out that we might as well treat each $Y_i$ as an unordered tuple. That is, for any $Y_i$, define
52 $X_i \in \Delta^n$ to be the vector of frequencies whose $a$-th entry is $\frac{1}{k} \sum_{j=1}^k \mathbb{1}[Y_i^j = a]$ for all $a \in [n]$. Then
53 for each, $i \in S_G$, $X_i$ is an independent draw from $\mathrm{Mul}_k(\mu_i)$. Henceforth, we will work solely in this
54 frequency vector perspective.

## 1.3 Elementary Facts

56 In this section we collect miscellaneous elementary facts that will be useful in subsequent sections.

57 **Fact 1.2.** *For $X_1, \cdots, X_m \in \mathbb{R}^n$, weights $w : [m] \to \mathbb{R}_{\geq 0}$, $v \in \mathbb{R}^n$, $\mu \in \mathbb{R}^n$, and $\Sigma \in \mathbb{R}^{n \times n}$*
58 *symmetric,*

$$\sum w_i \left\langle (X_i - \mu)^{\otimes 2}, \Sigma \right\rangle = \sum w_i \left\langle (X_i - \mu(w))^{\otimes 2}, \Sigma \right\rangle + \|w\|_1 \cdot \left\langle (\mu(w) - \mu)^{\otimes 2}, \Sigma \right\rangle. \quad (2)$$

59 *In particular, by taking $\Sigma = vv^\top$ for any $v \in \mathbb{R}^n$,*

$$\sum w_i \langle X_i - \mu, v \rangle^2 = \sum w_i \langle X_i - \mu(w), v \rangle^2 + \|w\|_1 \cdot \langle \mu(w) - \mu, v \rangle^2.$$

60 *That is, the function $\nu \mapsto \sum_i w_i \langle X_i - \nu, v \rangle^2$ is minimized over $\nu \in \mathbb{R}^n$ by $\nu = \mu(w)$.*

61 *Proof.* Without loss of generality we may assume $\|w\|_1 = 1$. Using the fact that $\langle u^{\otimes 2}, \Sigma \rangle -$
62 $\langle v^{\otimes 2}, \Sigma \rangle = (u - v)^\top \Sigma (u + v)$ for symmetric $\Sigma$, we see that

$$\left\langle (X_i - \mu)^{\otimes 2} - (X_i - \mu(w))^{\otimes 2}, \Sigma \right\rangle = (\mu(w) - \mu)^\top \Sigma (2X_i - \mu - \mu(w)).$$

63 Because $\sum w_i X_i = \mu(w)$, we see that

$$\sum w_i (\mu(w) - \mu)^\top \Sigma (2X_i - \mu - \mu(w)) = \left\langle (\mu(w) - \mu)^{\otimes 2}, \Sigma \right\rangle,$$

64 from which (2) follows. The remaining parts of the claim follow trivially. $\square$

65 **Fact 1.3.** *For any $0 < \epsilon < 1$, let weights $w : [N] \to [0, 1/N]$ satisfy $\sum_{i \in [N]} w_i \geq 1 - O(\epsilon)$. If $w'$*
66 *is the set of weights defined by $w_i' = w_i$ for $i \in S_G$ and $w_i' = 0$ otherwise, and if $|S_G| \geq (1 - \epsilon)N$,*
67 *then we have that $\|\mu(w) - \mu(w')\|_1 \leq O(\epsilon)$.*

68 *Proof.* We may write

$$\|\mu(w) - \mu(w')\|_1 \leq \|\frac{1}{\|w\|_1} \sum_{i \in S_B} w_i X_i\|_1 + \left( \frac{1}{\|w\|_1} - \frac{1}{\|w'\|_1} \right) \| \sum_{i \in S_G} w_i X_i \|_1$$

$$\leq O(\epsilon) + \left( \frac{1}{\|w\|_1} - \frac{1}{\|w'\|_1} \right) \| \sum_{i \in S_G} w_i X_i \|_1 \leq O(\epsilon),$$

69 where the first step follows by definition of $\mu(\cdot)$ and by triangle inequality, the second step follows by
70 the fact that $|S_B| \leq \epsilon N$, and the third step follows by the fact that $\|\|w\|_1 - \|w'\|_1\| = \left| \sum_{i \in S_B} w_i \right| \leq \epsilon$,
71 while $\| \sum_{i \in S_G} w_i X_i \|_1 \leq 1$ as the samples $X_i$ lie in $\Delta^n$. $\square$

It will be useful to have a basic bound on the Frobenius norm of $M(w, \nu)$.

**Lemma 1.4.** *For any $\nu \in \Delta^n$ and any weights $w$ for which $\sum w_i = 1$, we have that $\|M(w, \nu)\|_F \leq 3$.*

*Proof.* For any sample $X \in \Delta^n$, we have that

$$\|(X - \nu)(X - \nu)^\top\|_F \leq \|X - \nu\|_2^2 \leq 2$$

and

$$\|B(\nu)\|_F \leq \frac{1}{k}\|\nu\|_2 + \frac{1}{k}\|\nu\|_2^2 \leq 2/k,$$

from which the lemma follows by triangle inequality and the assumption that $\sum w_i = 1$. $\qquad\square$

## 1.4 $\mathcal{A}_K$ Norms and VC Complexity

In this section we review basics about learning distributions which are close to piecewise polynomial.

**Definition 1.5** ($\mathcal{A}_K$ norms, see e.g. [DL01])**.** *For positive integers $K \leq n$, define $\mathcal{A}_K$ to be the set of all unions of at most $K$ disjoint intervals over $[n]$, where an interval is any subset of $[n]$ of the form $\{a, a + 1, \cdots, b - 1, b\}$. The $\mathcal{A}_K$ distance between two distributions $\mu, \nu$ over $[n]$ is*

$$\|\mu - \nu\|_{\mathcal{A}_K} = \max_{S \in \mathcal{A}_K} |\mu(S) - \mu(S)|.$$

*Equivalently, say that $v \in \{\pm 1\}^n$ has $2K$ sign changes if there are exactly $2K$ indices $i \in [n - 1]$ for which $v_{i+1} \neq v_i$. Then if $\mathcal{V}_{2K}^n$ denotes the set of all such $v$, we have*

$$\|\mu - \nu\|_{\mathcal{A}_K} = \frac{1}{2} \max_{v \in \mathcal{V}_{2K}^n} \langle \mu - \nu, v \rangle.$$

*Note that*

$$\| \cdot \|_{\mathcal{A}_1} \leq \| \cdot \|_{\mathcal{A}_2} \leq \cdots \leq \| \cdot \|_{\mathcal{A}_{n/2}} = \| \cdot \|_{TV}.$$

**Definition 1.6.** *We say that a distribution $\mu$ over $[n]$ is $(\eta, s)$-piecewise degree-$d$ if there is a partition of $[n]$ into $t$ disjoint intervals $\{[a_i, b_i]\}_{1 \leq i \leq t}$, together with univariate degree-$d$ polynomials $r_1, \cdots, r_t$ and a distribution $\mu'$ on $[n]$, such that $d_{TV}(\mu, \mu') \leq \eta$ and such that for all $i \in [t]$, $\mu'(x) = r_i(x)$ for all $x \in [n]$ in $[a_i, b_i]$.*

A proof of the following lemma, a consequence of [ADLS17], can be found in [CLM19].

**Lemma 1.7** (Lemma 5.1 in [CLM19], follows by [ADLS17])**.** *Let $K = s(d + 1)$. If $\mu$ is $(\eta, s)$-piecewise degree-$d$ and $\|\mu - \hat{\mu}\|_{\mathcal{A}_K} \leq \zeta$, then there is an algorithm which, given the vector $\hat{\mu}$, outputs a distribution $\mu^*$ for which $d_{TV}(\mu, \mu^*) \leq 2\zeta + 4\eta$ in time $\text{poly}(s, d, 1/\eta)$.*

Henceforth, we will focus solely on the problem of learning in $\mathcal{A}_\ell$ norm, where

$$\ell \triangleq 2s(d + 1). \tag{3}$$

## 1.5 Haar Wavelets

We briefly recall the definition of Haar wavelets, further details and examples of which can be found in [CLM19].

**Definition 1.8.** *Let $m$ be a positive integer and let $n = 2^m$. The Haar wavelet basis is an orthonormal basis over $\mathbb{R}^n$ consisting of the father wavelet $\psi_{0_{father},0} = n^{-1/2} \cdot \mathbf{1}$, the mother wavelet $\psi_{0_{mother},0} = n^{-1/2} \cdot (1, \cdots, 1, -1, \cdots, -1)$ (where $(1, \cdots, 1, -1, \cdots, -1)$ contains $n/2$ 1's and $n/2$ -1's), and for every $i, j$ for which $1 \leq i < m$ and $0 \leq j < 2^i$, the wavelet $\psi_{i,j}$ whose $2^{m-i} \cdot j + 1, \cdots, 2^{m-i} \cdot j + 2^{m-i-1}$-th coordinates are $2^{-(m-i)/2}$ and whose $2^{m-i} \cdot j + (2^{m-i-1} + 1), \cdots, 2^{m-i} \cdot j + 2^{m-i}$-th coordinates are $-2^{-(m-i)/2}$, and whose remaining coordinates are 0.*

Additionally, we will use the following notation when referring to Haar wavelets:

- Let $H_m$ denote the $n \times n$ matrix whose rows consist of the vectors of the Haar wavelet basis for $\mathbb{R}^n$. When the context is clear, we will omit the subscript and refer to this matrix as $H$.

- For $\nu \in [n]$, if the $\nu$-th element of the Haar wavelet basis for $\mathbb{R}^n$ is some $\psi_{i,j}$, then define the weight $\mathbf{h}^{(\nu)} \triangleq 2^{-(m-i)/2}$.

- For any index $i \in \{0_{\text{father}}, 0_{\text{mother}}, 1, \cdots, m-1\}$, let $T_i \subset [n]$ denote the set of indices $\nu$ for which the $\nu$-th Haar wavelet is of the form $\psi_{i,j}$ for some $j$.

- Given any $p \geq 1$, define the *Haar-weighted $L^p$ norm* $\|\cdot\|_{p;\mathbf{h}}$ on $\mathbb{R}^n$ by $\|w\|_{p;\mathbf{h}} \triangleq \|w'\|_p$, where for every $a \in [n]$, $w'_a \triangleq \mathbf{h}^{(a)} w_a$. Likewise, given any norm $\|\cdot\|_*$ on $\mathbb{R}^{n \times n}$, define the Haar-weighted $*$-norm $\|\cdot\|_{*;\mathbf{h}}$ on $\mathbb{R}^{n \times n}$ by $\|\mathbf{M}\|_{*;\mathbf{h}} \triangleq \|\mathbf{M}'\|_*$, where for every $a, b \in [n]$, $\mathbf{M}'_{a,b} \triangleq \mathbf{h}^{(a)} \mathbf{h}^{(b)} \mathbf{M}_{a,b}$.

The key observation is that any $v \in \{\pm 1\}^n$ with at most $\ell$ sign changes, where $\ell$ is given by (3), has an $(\ell \log n + 1)$-sparse representation in the Haar wavelet basis. We will use the following fundamental fact about Haar wavelets, part of which appears as Lemma 6.3 in [CLM19].

**Lemma 1.9.** *Let $v \in \{\pm 1\}^n$ have at most $\ell$ sign changes. Then $Hv$ has at most $\ell \log n + 1$ nonzero entries, and furthermore $\|Hv\|_{\infty;\mathbf{h}} \leq 1$. In particular, $\|Hv\|_{2;\mathbf{h}}^2, \|Hv\|_{1;\mathbf{h}} \leq \ell \log n + 1$.*

*Proof.* We first show that $Hv$ has at most $\ell \log n + 1$ nonzero entries. For any $\psi_{i,j}$ with nonzero entries at indices $[a, b] \subset [n]$ and such that $i \neq 0_{\text{father}}$, if $v$ has no sign change in the interval $[a, b]$, then $\langle \psi_{i,j}, v \rangle = 0$. For every index $\nu \in [n]$ at which $v$ has a sign change, there are at most $m = \log n$ choices of $i, j$ for which $\psi_{i,j}$ has a nonzero entry at index $\nu$, from which the claim follows by a union bound over all $\ell$ choices of $\nu$, together with the fact that $\langle \psi_{0_{\text{father}},0}, v \rangle$ may be nonzero.

Now for each $(i, j)$ for which $\langle \psi_{i,j}, v \rangle \neq 0$, note that

$$2^{-(m-i)/2} \cdot |\langle \psi_{i,j}, v \rangle| \leq 2^{-(m-i)/2} \cdot \left( 2^{-(m-i)/2} \cdot 2^{m-i} \right) = 1,$$

as claimed. The bounds on $\|Hv\|_{1;\mathbf{h}}, \|Hv\|_{2;\mathbf{h}}^2$ follow immediately. $\qquad\square$

## 2 SDP for Finding the Direction of Largest Variance

Recall that in [JO19], the authors consider the binary optimization problem $\max_{v \in \{0,1\}^n} |v^\top M_U v|$. We would like to approximate the optimization problem $\max_{v \in \mathcal{V}_\ell^n} |v^\top M_U v|$. Motivated by [CLM19] and Lemma 1.9, we consider the following convex relaxation:

**Definition 2.1.** *Let $\ell$ be given by (3). Let $\mathcal{K}$ denote the (convex) set of all matrices $\Sigma \in \mathbb{R}^{n \times n}$ for which*

1. $\|\Sigma\|_{\max} \leq 1$.

2. $\|H \Sigma H^\top\|_{1,1;\mathbf{h}} \leq \ell \log n + 1$.

3. $\|H \Sigma H^\top\|_{F;\mathbf{h}}^2 \leq \ell \log n + 1$.

4. $\|H \Sigma H^\top\|_{\max;\mathbf{h}} \leq 1$.

5. $\Sigma \succeq 0$.

*Let $\|\cdot\|_\mathcal{K}$ denote the associated norm given by $\|\mathbf{M}\|_\mathcal{K} \triangleq \sup_{\Sigma \in \mathcal{K}} |\langle \mathbf{M}, \Sigma \rangle|$. By abuse of notation, for vectors $v \in \mathbb{R}^n$ we will also use $\|v\|_\mathcal{K}$ to denote $\|vv^\top\|_\mathcal{K}^{1/2}$.*

*Because $\mathcal{K}$ has an efficient separation oracle, one can compute $\|\cdot\|_\mathcal{K}$ in polynomial time.*

**Remark 2.2.** *Note that, besides not being a sum-of-squares program like the one considered in [CLM19], this relaxation is also slightly different because of Constraints 3 and 4. As we will see in Section B, these additional constraints will be crucial for getting refined sample complexity bounds.*

Note that Lemma 1.9 immediately implies that $\mathcal{K}$ is a relaxation of $\mathcal{V}_\ell^n$:

**Corollary 2.3** (Corollary of Lemma 1.9). *$vv^\top \in \mathcal{K}$ for any $v \in \mathcal{V}_\ell^n$.*

Note also that Constraint 1 in Definition 2.1 ensures that $\|\cdot\|_\mathcal{K}$ is weaker than $\|\cdot\|_1$ and more generally that:

**Fact 2.4.** *For any $a, b \in \mathbb{R}^n$ and $\Sigma \in \mathcal{K}$, $a^\top \cdot \Sigma \cdot b \leq \|a\|_1 \cdot \|b\|_1$. In particular, for any $v \in \mathbb{R}^n$, $\|v\|_{\mathcal{K}} \leq \|v\|_1$.*

As a consequence, we conclude the following useful fact about stability of the $B(\cdot)$ matrix.

**Corollary 2.5.** *For any $\mu, \mu' \in \Delta^n$, $\|B(\mu) - B(\mu')\|_{\mathcal{K}} \leq \frac{3}{k} \|\mu - \mu'\|_1$.*

*Proof.* Take any $\Sigma \in \mathcal{K}$. By symmetry, it is enough to show that $\langle B(\mu) - \mathcal{B}(\mu'), \Sigma \rangle \leq \frac{3}{k} \|\mu - \mu'\|_1$. By Constraint 1, we have that $\langle \mu - \mu', \mathrm{diag}(\Sigma) \rangle \leq \|\mu - \mu'\|_1$. On the other hand, note that

$$\mu'^\top \Sigma \mu' - \mu^\top \Sigma \mu = (\mu' - \mu)^\top \Sigma (\mu' + \mu) \leq \|\mu' - \mu\|_1 \cdot \|\mu' + \mu\|_1 \leq 2\|\mu' - \mu\|_1,$$

where the second step follows from Fact 2.4. The corollary now follows. $\qquad\square$

Note that if the solution to the convex program $\mathrm{argmax}_{\Sigma \in \mathcal{K}} \langle M_U, \Sigma \rangle$ were actually integral, that is, some rank-1 matrix $vv^\top$ for $v \in \mathcal{V}_\ell^n$, it would correspond to the direction $v$ in which the samples in $U$ have the largest discrepancy between the empirical variance and the variance predicted by the empirical mean. Then $v$ would correspond to a subset of the domain $[s]$ on which one could filter out bad points as in [JO19]. In the sequel, we will show that this kind of analysis applies *even if the solution to $\mathrm{argmax}_{\Sigma \in \mathcal{K}} \langle M_U, \Sigma \rangle$ is not integral*.

# 3 Filtering Algorithm and Analysis

In this section we prove our main theorem, stated formally below:

**Theorem 3.1.** *Let $\mu$ be an $(\eta, s)$-piecewise degree-d distribution over $[n]$. Then for any $0 < \epsilon < 1/2$ smaller than some absolute constant, and any $0 < \delta < 1$, there is a $\mathrm{poly}(n, k, 1/\epsilon, 1/\delta)$-time algorithm LEARNWITHFILTER which, given*

$$N = \widetilde{O}\left(\log(1/\delta)(s^2 d^2/\epsilon^2)\log^3(n)\right),$$

*$\epsilon$-corrupted, $\omega$-diverse batches of size $k$ from $\mu$, outputs an estimate $\hat{\mu}$ such that $\|\hat{\mu} - \mu\|_1 \leq O\left(\eta + \omega + \frac{\epsilon\sqrt{\log 1/\epsilon}}{\sqrt{k}}\right)$ with probability at least $1 - \delta$ over the samples.*

In Section 3.1, we first describe and prove guarantees for a basic but important subroutine, 1DFILTER, of our algorithm. In Section 3.2, we describe our learning algorithm, LEARNWITHFILTER, in full. In Section 3.3 we define the deterministic conditions that the dataset must satisfy for LEARNWITHFILTER to succeed, deferring the proof that these deterministic conditions hold with high probability (Lemma 3.6) to Appendix A. In Section 3.4 we prove a key geometric lemma (Lemma 3.7). Finally, in Section 3.5, we complete the proof of correctness of LEARNWITHFILTER.

## 3.1 Univariate Filter

In this section, we define and analyze a simple deterministic subroutine 1DFILTER which takes as input a set of weights $w$ and a set of scores on the batches $X_1, \cdots, X_N$, and outputs a new set of weights $w'$ such that, if the weighted average of the scores among the bad batches exceeds that of the scores among the good batches, then $w'$ places even less weight relatively on the bad batches than does $w$. This subroutine is given in Algorithm 1 below.

---
**Algorithm 1:** 1DFILTER$(\tau, w)$

**Input:** Scores $\tau : [N] \to \mathbb{R}_{\geq 0}$, weights $w : [N] \to \mathbb{R}_{\geq 0}$
**Output:** New weights $w'$ with even less mass on bad points than good points (see Lemma 3.2)
1   $\tau_{\max} \leftarrow \max_{i:w_i > 0} \tau_i$
2   $w_i' \leftarrow \left(1 - \frac{\tau_i}{\tau_{\max}}\right) w_i$ for all $i \in [N]$
3   Output $w'$

---

**Lemma 3.2.** *Let $\tau : [N] \to \mathbb{R}_{\geq 0}$ be a set of scores, and let $w : [N] \to \mathbb{R}_{\geq 0}$ be a weight. Given a partition $[N] = S_G \sqcup S_B$ for which*

$$\sum_{i \in S_G} w_i \tau_i < \sum_{i \in S_B} w_i \tau_i,$$

*then the output $w'$ of $1$DFILTER$(\tau, w)$ satisfies $(a)$ $w'_i \leq w_i$ for all $i \in [N]$, $(b)$ the support of $w'$ is a strict subset of the support of $w$, and $(c)$ $\sum_{i \in S_G} w_i - w'_i < \sum_{i \in S_B} w_i - w'_i$.*

*Proof.* $(a)$ and $(b)$ are immediate. For $(c)$, note that

$$\sum_{i \in S_G} w_i - w'_i = \frac{1}{\tau_{\max}} \sum_{i \in S_G} \tau_i w_i < \frac{1}{\tau_{\max}} \sum_{i \in S_B} \tau_i w_i = \sum_{i \in S_B} w_i - w'_i,$$

from which the lemma follows. $\qquad\qquad\qquad\qquad\qquad\qquad\qquad\qquad\qquad\qquad\qquad\quad$ $\square$

We note that this kind of downweighting scheme and its analysis are not new, see e.g. Lemma 4.5 from [CSV17] or Lemma 17 from [SCV18].

## 3.2 Algorithm Specification

We can now describe our algorithm LEARNWITHFILTER. At a high level, we maintain weights $w : [N] \to \mathbb{R}_{\geq 0}$ for each of the batches. In every iteration, we compute $\Sigma \in \mathcal{K}$ maximizing $|\langle M(w), \Sigma \rangle|$. If $|\langle M(w), \Sigma \rangle| \leq O\left(\frac{\epsilon}{k} \log 1/\epsilon\right)$, then output $\mu(w)$. Otherwise, update the weights as follows: for every batch $X_i$, compute the score $\tau_i$ given by

$$\tau_i \triangleq \left\langle (X_i - \mu(w))^{\otimes 2}, \Sigma \right\rangle, \tag{4}$$

and set the weights to be the output of $1$DFILTER$(\tau, w)$. The pseudocode for LEARNWITHFILTER is given in Algorithm 2 below.

---

**Algorithm 2:** LEARNWITHFILTER$(\{X_i\}_{i \in [N]}, \epsilon)$

---

**Input:** Frequency vectors $X_1, \cdots, X_N$ coming from an $\epsilon$-corrupted, $\omega$-diverse set of batches from $\mu$, where $\mu$ is $(\eta, s)$-piecewise, degree $d$

**Output:** $\hat{\mu}$ such that $\|\hat{\mu} - \mu\|_1 \leq O\left(\eta + \omega + \frac{\epsilon\sqrt{\log 1/\epsilon}}{\sqrt{k}}\right)$, provided uncorrupted samples $\epsilon$-good

1   $w \leftarrow w([N])$
2   **while** $\|M(w)\|_{\mathcal{K}} \geq \Omega(\omega + \frac{\epsilon}{k} \log 1/\epsilon)$ **do**
3      $\Sigma \leftarrow \mathrm{argmax}_{\Sigma' \in \mathcal{K}} |\langle M(w), \Sigma \rangle|$
4      Compute scores $\tau : [N] \to \mathbb{R}_{\geq 0}$ according to (4).
5      $w \leftarrow 1$DFILTER$(\tau, w)$
6   Using the algorithm of [ADLS17] (see Lemma 1.7), output the $s$-piecewise, degree-$d$ distribution $\hat{w}$ minimizing $\|\mu(w) - \hat{\mu}\|_{s(d+1)}$ (up to additive error $\eta$).

---

## 3.3 Deterministic Condition

**Definition 3.3** ($\epsilon$-goodness). *Take a set of points $U \subset [N]$, and let $\{\mu_i\}_{i \in U}$ be a collection of distributions over $[n]$. For any $W \subseteq U$, define $\overline{\mu}_W \triangleq \frac{1}{|W|} \sum_{i \in W} \mu_i$. Denote $\overline{\mu} \triangleq \overline{\mu}_U$.*

*We say $U$ is $\epsilon$-good if it satisfies that for all $W \subset U$ for which $|W| = \epsilon |U|$,*

*(I) (Concentration of mean)*

$$\|\mu(U) - \overline{\mu}\|_{\mathcal{K}} \leq O\left(\frac{\epsilon\sqrt{\log 1/\epsilon}}{\sqrt{k}}\right) \quad and \quad \|\mu(W) - \overline{\mu}_W\|_{\mathcal{K}} \leq O\left(\frac{\sqrt{\log 1/\epsilon}}{\sqrt{k}}\right)$$

*(II) (Concentration of covariance)*

$$\|M(\hat{w}(U), \{\mu_i\}_{i \in U})\|_{\mathcal{K}} \leq O\left(\frac{\epsilon \log 1/\epsilon}{k}\right) \quad and \quad \|A(\hat{w}(W), \{\mu_i\}_{i \in W})\|_{\mathcal{K}} \leq O\left(\frac{\log 1/\epsilon}{k}\right)$$

*(III) (Concentration of variance proxy)*

$$\|B(\hat{\mu}(U)) - B(\{\mu_i\}_{i \in U})\|_{\mathcal{K}} \leq O(\omega^2/k + \epsilon/k)$$

*(IV) (Heterogeneity has negligible effect, see Lemma 3.4)*

$$\sup_{\Sigma \in \mathcal{K}} \left\{ \frac{1}{|U|} \sum_{i \in U} (\mu_i - \overline{\mu})^\top \cdot \Sigma \cdot (X_i - \mu_i) \right\} \leq O\left( \omega \cdot \frac{\epsilon \sqrt{\log 1/\epsilon}}{\sqrt{k}} \right).$$

$$\sup_{\Sigma \in \mathcal{K}} \left\{ \frac{1}{|W|} \sum_{i \in W} (\mu_i - \overline{\mu})^\top \cdot \Sigma \cdot (X_i - \mu_i) \right\} \leq O\left( \omega \cdot \frac{\sqrt{\log 1/\epsilon}}{\sqrt{k}} \right).$$

We first remark that we only need extremely mild concentration in Condition (III), but it turns out this suffices in the one place where we use it (see Lemma 3.9).

Additionally, note that we can completely ignore Condition (IV) when $\omega = 0$. The following makes clear why it is useful when $\omega > 0$.

**Lemma 3.4.** *For $\epsilon$-good $U$, all $W \subset U$ of size $\epsilon|U|$, and all $\Sigma \in \mathcal{K}$,*

$$\|A(\hat{\mu}(U), \overline{\mu}) - A(\hat{\mu}(U), \{\mu_i\})\|_{\mathcal{K}} \leq O\left( \omega + \frac{\epsilon \sqrt{\log 1/\epsilon}}{\sqrt{k}} \right)^2$$

$$\|A(\hat{\mu}(W), \overline{\mu}) - A(\hat{\mu}(W), \{\mu_i\})\|_{\mathcal{K}} \leq O\left( \omega + \frac{\sqrt{\log 1/\epsilon}}{\sqrt{k}} \right)^2.$$

*Proof.* For $S = U$ or $S = W$ and any $\Sigma \in \mathcal{K}$,

$$\langle \Sigma, A(\hat{\mu}(S), \overline{\mu}) - A(\hat{\mu}(S), \{\mu_i\}) \rangle$$

$$= \frac{1}{|S|} \sum_{i \in S} \langle (X_i - \overline{\mu})^{\otimes 2} - (X_i - \mu_i)^{\otimes 2}, \Sigma \rangle$$

$$= \frac{1}{|S|} \sum_{i \in S} (\mu_i - \overline{\mu})^\top \cdot \Sigma \cdot (2X_i - \mu_i - \overline{\mu})$$

$$= \frac{2}{|S|} \sum_{i \in S} (\mu_i - \overline{\mu})^\top \cdot \Sigma \cdot (X_i - \mu_i) + \frac{1}{|S|} \sum_{i \in S} \langle (\mu_i - \overline{\mu})^{\otimes 2}, \Sigma \rangle. \tag{5}$$

The first (resp. second) part of the lemma follows by taking $S = U$ (resp. $S = W$) and invoking the first (resp. second) part of Condition (IV) of $\epsilon$-goodness to upper bound the first term in (5), and Fact 2.4 and the fact that $\|\mu_i - \overline{\mu}\|_1 \leq \omega$ for all $i$ to upper bound the second term in (5). $\square$

**Corollary 3.5.** *If $U$ is $\epsilon$-good and $\overline{\mu} \triangleq \frac{1}{|U|} \sum_{i \in U} \mu_i$, then*

$$\|A(\hat{w}(U), \overline{\mu}) - B(\{\mu_i\})\|_{\mathcal{K}} \leq O\left( \omega + \frac{\epsilon \sqrt{\log 1/\epsilon}}{\sqrt{k}} \right)^2.$$

*Proof.* This follows immediately from Lemma 3.4 and the first part of Condition (II) of $\epsilon$-goodness. $\square$

In Appendix A, we will show that for $N$ sufficiently large, the set $S_G$ of uncorrupted batches will satisfy the above deterministic condition.

**Lemma 3.6** (Regularity of good samples). *If $U$ is a set of $\widetilde{\Omega}\left(\log(1/\delta)(\ell^2/\epsilon^2) \cdot \log^3(n)\right)$ independent samples from $Mul_k(\mu_1), ..., Mul_k(\mu_{|U|})$, then $U$ is $\epsilon$-good with probability at least $1 - \delta$.*

## 3.4 Key Geometric Lemma

The key property of $\epsilon$-good sets is the following geometric lemma bounding the accuracy of an estimate $\mu(w)$ given by weights $w$ in terms of $\|M(w)\|_{\mathcal{K}}$.

**Lemma 3.7** (Spectral signatures). *If $S_G$ is $\epsilon$-good and $|S_G| \geq (1 - \epsilon)N$, then for any $w \in \mathcal{W}_\epsilon$,*

$$\|\mu(w) - \overline{\mu}\|_{\mathcal{K}} \leq O\left(\frac{\epsilon}{\sqrt{k}}\sqrt{\log 1/\epsilon} + \epsilon \cdot \omega + \sqrt{\epsilon\left(\|M(w)\|_{\mathcal{K}} + \omega^2 + \frac{\epsilon}{k}\log 1/\epsilon\right)}\right).$$

It turns out the proof ingredients for Lemma 3.7 will also be useful in our analysis of LEARNWITH-FILTER later, so we will now prove this lemma in full.

*Proof.* Take any $\Sigma \in \mathcal{K}$. Recalling that $\Sigma$ is psd by Constraint 5 in Definition 2.1, we will sometimes write it as $\Sigma = \mathbb{E}_v[vv^\top]$, where the distribution over $v$ is defined according to the eigendecomposition of $\Sigma$. We wish to bound $\mathbb{E}_v\left[\langle\mu(w) - \mu, v\rangle^2\right]$. By splitting $w_i \triangleq 1/N - \delta_i$ for $i \in S_G$, we have that

$$\langle\mu(w) - \overline{\mu}, v\rangle = \sum_{i=1}^N w_i\langle X_i - \overline{\mu}, v\rangle$$

$$= \left\langle\frac{|S_G|}{N}(\mu(S_G) - \overline{\mu}), v\right\rangle - \sum_{i \in S_G}\delta_i\langle X_i - \overline{\mu}, v\rangle + \sum_{i \in S_B}w_i\langle X_i - \overline{\mu}, v\rangle,$$

$$= \left\langle\frac{|S_G|}{N}(\mu(S_G) - \overline{\mu}), v\right\rangle - \sum_{i \in S_G}\delta_i\langle X_i - \overline{\mu}, v\rangle + \sum_{i \in S_B}w_i\langle X_i - \mu(w), v\rangle + \langle\mu(w) - \overline{\mu}, v\rangle\sum_{i \in S_B}w_i.$$

We may rewrite this as

$$\left(1 - \sum_{i \in S_B}w_i\right)\langle\overline{\mu}(w) - \mu, v\rangle = \left\langle\frac{|S_G|}{N}(\mu(S_G) - \overline{\mu}), v\right\rangle - \sum_{i \in S_G}\delta_i\langle X_i - \overline{\mu}, v\rangle + \sum_{i \in S_B}w_i\langle X_i - \overline{\mu}(w), v\rangle.$$

Note further that

$$\sum_{i \in S_G}\delta_i\langle X_i - \overline{\mu}, v\rangle = \sum_{i \in S_G}\delta_i\langle X_i - \mu_i, v\rangle + \sum_{i \in S_G}\delta_i\langle\mu_i - \overline{\mu}, v\rangle,$$

so in particular,

$$\frac{1}{4}\left(1 - \sum_{i \in S_B}w_i\right)^2 \cdot \mathbb{E}_v\left[\langle\mu(w) - \overline{\mu}, v\rangle^2\right] \leq ① + ② + ③ + ④ \tag{6}$$

where

$$① \triangleq \frac{|S_G|^2}{N^2}\mathbb{E}_v\left[\langle\mu(S_G) - \overline{\mu}, v\rangle^2\right] \qquad ② \triangleq \mathbb{E}_v\left[\left(\sum_{i \in S_G}\delta_i\langle X_i - \mu_i, v\rangle\right)^2\right]$$

$$③ \triangleq \mathbb{E}_v\left[\left(\sum_{i \in S_G}\delta_i\langle\mu_i - \mu, v\rangle\right)^2\right] \qquad ④ \triangleq \mathbb{E}_v\left[\left(\sum_{i \in S_B}w_i\langle X_i - \mu(w), v\rangle\right)^2\right]$$

For ①, note that

$$① \leq \frac{|S_G|^2}{N^2}\|\mu(S_G) - \overline{\mu}\|_{\mathcal{K}}^2 \leq O\left(\frac{\epsilon^2\log 1/\epsilon}{k}\right)$$

243    by the first part of Condition (I) of $\epsilon$-goodness of $S_G$ and the fact that $|S_G|/N \geq 1 - \epsilon$.

244    For ②, by Cauchy-Schwarz we have that

$$
\begin{aligned}
② &\leq \left( \sum_{i \in S_G} \delta_i \right) \cdot \mathbb{E}_v \left[ \sum_{i \in S_G} \delta_i \langle X_i - \mu_i, v \rangle^2 \right] \\
&\leq \epsilon \cdot \left\langle \sum_{i \in S_G} \delta_i (X_i - \mu_i)^{\otimes 2}, \mathbb{E}_v[vv^\top] \right\rangle \\
&= \epsilon \left\langle A(\delta, \{\mu_i\}), \Sigma \right\rangle \\
&\leq O\left( \frac{\epsilon^2}{k} \log 1/\epsilon \right),
\end{aligned}
\tag{7}
$$

245    where the last step follows by Lemma 3.8 below.

246    For ③, again by Cauchy-Schwarz,

$$
\begin{aligned}
③ &\leq \left( \sum_{i \in S_G} \delta_i \right) \cdot \mathbb{E}_v \left[ \sum_{i \in S_G} \delta_i \langle \mu_i - \mu, v \rangle^2 \right] \\
&\leq \epsilon \cdot \sum_{i \in S_G} \delta_i \|\mu_i - \mu\|_{\mathcal{K}}^2 \\
&\leq \epsilon^2 \cdot \max_{i \in S_G} \|\mu_i - \mu\|_1^2 \\
&\leq \epsilon^2 \cdot \omega^2,
\end{aligned}
$$

247    where the penultimate step follows by Fact 2.4.

248    Finally, we will relate ④ to $\|M(w)\|_{\mathcal{K}}$. Let $w'$ be the set of weights given by $w'_i = w_i$ for $i \in S_G$
249    and $w'_i = 0$ for $i \notin S_G$. By another application of Cauchy-Schwarz,

$$
\begin{aligned}
④ &\leq \left( \sum_{i \in S_B} w_i \right) \cdot \mathbb{E}_v \left[ \sum_{i \in S_B} w_i \langle X_i - \mu(w), v \rangle^2 \right] \\
&\leq \epsilon \left( \mathbb{E}_v \left[ \sum_{i=1}^{N} w_i \langle X_i - \mu(w), v \rangle^2 \right] - \mathbb{E}_v \left[ \sum_{i \in S_G} w_i \langle X_i - \mu(w), v \rangle^2 \right] \right) \\
&= \epsilon \left\langle A(w, \mu(w)) - A(w', \mu(w)), \Sigma \right\rangle \tag{8} \\
&\leq \epsilon \left\langle A(w, \mu(w)) - A(w', \mu(w')), \Sigma \right\rangle \tag{9} \\
&\leq \epsilon \left\langle A(w, \mu(w)) - \frac{1}{\sum w'_i} B(\mu(w')), \Sigma \right\rangle + O\left( \epsilon \cdot \omega^2 + \frac{\epsilon^2}{k} \log 1/\epsilon \right) \tag{10} \\
&= \epsilon \langle M(w), \Sigma \rangle + \epsilon \left\langle B(\mu(w)) - \frac{1}{\sum w'_i} B(\mu(w')), \Sigma \right\rangle + O\left( \epsilon \cdot \omega^2 + \frac{\epsilon^2}{k} \log 1/\epsilon \right) \\
&\leq \epsilon \|M(w)\|_{\mathcal{K}} + \epsilon \|B(\mu(w)) - \frac{1}{\sum w'_i} B(\mu(w'))\|_{\mathcal{K}} + O\left( \epsilon \cdot \omega^2 + \frac{\epsilon^2}{k} \log 1/\epsilon \right) \tag{11}
\end{aligned}
$$

250    where (8) follows by the definition of $A(w, \nu)$, (9) follows by Fact 1.2, (10) follows by Lemma 3.9
251    below. Lastly, by triangle inequality, we may upper bound $\|B(\mu(w)) - \frac{1}{\sum w'_i} B(\mu(w'))\|_{\mathcal{K}}$ by

$$
\|B(\mu(w)) - B(\mu(w'))\|_{\mathcal{K}} + O(\epsilon) \cdot \|B(\mu(w'))\|_{\mathcal{K}} \leq \frac{3}{k} \|\mu(w) - \mu(w')\|_1 + O(\epsilon/k) \leq O(\epsilon/k), \tag{12}
$$

252    where the first inequality follows by Corollary 2.5, and the bound on $\|\mu(w) - \mu(w')\|_1$ in the last
253    step follows from Fact 1.3. The lemma then follows from (6), (7), (11), and (12).    □

254    Next, we show in Lemma 3.8 that small subsets of the good samples cannot contribute too much to
255    the total energy. Lemma 3.9, which bounds the norm of $M(w)$ for any set of weights $w$ which is
256    close to the uniform set of weights over $S_G$, will follow as a consequence.

**Lemma 3.8.** *For any $0 < \epsilon < 1/2$, if $U$ is $\epsilon$-good, and $\delta : U \to [0, 1/|U|]$ is a set of weights satisfying $\sum_{i \in U} \delta_i \leq \epsilon$, then we have the following bounds:*

1. $\|A(\delta, \{\mu_i\})\|_{\mathcal{K}} \leq O(\frac{\epsilon}{k} \log 1/\epsilon)$

2. $\|\sum_{i \in U} \delta_i(X_i - \mu_i)\|_{\mathcal{K}} \leq O(\frac{\epsilon}{\sqrt{k}}\sqrt{\log 1/\epsilon})$

3. $\|A(\delta, \overline{\mu})\|_{\mathcal{K}} \leq O\left(\epsilon \cdot \omega^2 + \frac{\epsilon \log 1/\epsilon}{k}\right)$

4. $\|\sum_{i \in U} \delta_i(X_i - \overline{\mu})\|_{\mathcal{K}} \leq O(\frac{\epsilon}{\sqrt{k}}\sqrt{\log 1/\epsilon} + \epsilon \cdot \omega).$

*Proof.* For the first part, we may assume without loss of generality that $\sum_{i \in U} \delta_i = \epsilon$. But then we may write $\delta$ as $\epsilon \, \mathbb{E}_W[\hat{w}(W)]$ for some distribution over subsets $W \subset U$ of size $\epsilon|U|$. By Jensen's inequality and the second part of Condition (II) of $\epsilon$-goodness of $U$, we conclude that

$$A(\delta, \{\mu_i\}) \leq \epsilon \cdot \mathbb{E}_W\left[\|A(\hat{w}(W), \{\mu_i\})\|_{\mathcal{K}}\right] \leq O\left(\frac{\epsilon}{k} \log 1/\epsilon\right),$$

giving the first part of the lemma.

For the second part, for any $\Sigma \in \mathcal{K}$ of the form $\Sigma = \mathbb{E}[vv^\top]$,

$$\left\langle \Sigma, \left(\sum_{i \in U} \delta_i(X_i - \mu_i)\right)^{\otimes 2} \right\rangle = \mathbb{E}\left[\left(\sum_{i \in U} \delta_i \langle X_i - \mu_i, v\rangle\right)^2\right]$$

$$\leq \mathbb{E}\left[\left(\sum_{i \in U} \delta_i\right) \cdot \left(\sum_{i \in U} \delta_i \langle X_i - \mu_i, v\rangle^2\right)\right]$$

$$\leq \epsilon \|A(\delta, \{\mu_i\})\| \leq O\left(\frac{\epsilon^2}{k} \log 1/\epsilon\right),$$

where the second step follows by Cauchy-Schwarz, the fourth step follows by the first part of the lemma. As this holds for all $\Sigma \in \mathcal{K}$, we get the second part of the lemma.

This also implies the fourth part of the lemma because

$$\|\sum_{i \in U} \delta_i(X_i - \overline{\mu})\|_{\mathcal{K}} \leq \|\sum_{i \in U} \delta_i(X_i - \mu_i)\|_{\mathcal{K}} + \|\sum_{i \in U} \delta_i(\mu_i - \overline{\mu})\|_{\mathcal{K}}$$

$$\leq O\left(\frac{\epsilon}{\sqrt{k}}\sqrt{\log 1/\epsilon}\right) + \sum_{i \in U} \delta_i\|\mu_i - \overline{\mu}\|_1$$

$$\leq O\left(\frac{\epsilon}{\sqrt{k}}\sqrt{\log 1/\epsilon} + \epsilon \cdot \omega\right),$$

where the second step follows by the above together with Fact 2.4 and triangle inequality.

Finally, for the third part of the lemma, upon regarding the weights $\delta$ as $\epsilon \, \mathbb{E}_W[\hat{w}(W)]$ as before and applying Jensen's to the second part of Lemma 3.4, we get that

$$\|A(\delta, \overline{\mu}) - A(\delta, \{\mu_i\})\|_{\mathcal{K}} \leq \epsilon \cdot O\left(\omega + \frac{\sqrt{\log 1/\epsilon}}{\sqrt{k}}\right)^2 \leq O\left(\epsilon \cdot \omega^2 + \frac{\epsilon \log 1/\epsilon}{k}\right).$$

The third part of the lemma then follows by the first part, together with triangle inequality. $\square$

**Lemma 3.9.** *If $S_G$ is $\epsilon$-good, and $w : S_G \to [0, 1]$ satisfies $\|w - \hat{w}(S_G)\|_1 \leq \epsilon$ and $\sum_{i \in S_G} w_i = 1$, then $\|M(w)\|_{\mathcal{K}} \leq O(\omega^2 + \frac{\epsilon}{k} \log 1/\epsilon).$*

*Proof.* Define $\delta_i = 1/|S_G| - w_i$ for all $i \in S_G$ and take any $\Sigma \in \mathcal{K}$.

By Fact 1.2 and the assumption that $\|w\|_1 = 1$,

$$\langle A(w, \mu(w)), \Sigma\rangle = \langle A(w, \overline{\mu}), \Sigma\rangle - \|\mu(w) - \overline{\mu}\|_{\mathcal{K}}^2. \tag{13}$$

For the second term on the right-hand side of (13), note that we can write

$$\mu(w) - \overline{\mu} = \sum_{i \in S_G} w_i(X_i - \overline{\mu})$$

$$= \sum_{i \in S_G} (1/|S_G| - \delta_i)(X_i - \overline{\mu})$$

$$= (\mu(S_G) - \overline{\mu}) - \sum_{i \in S_G} \delta_i(X_i - \overline{\mu})$$

$$= (\mu(S_G) - \overline{\mu}) - \sum_{i \in S_G} \delta_i(X_i - \mu_i) - \sum_{i \in S_G} \delta_i(\mu_i - \overline{\mu}),$$

where the first step follows by the fact that $\sum_{i \in S_G} w_i = 1$. So by triangle inequality,

$$\|\mu(w) - \overline{\mu}\|_{\mathcal{K}} \leq \|\mu(S_G) - \overline{\mu}\|_{\mathcal{K}} + \|\sum_{i \in S_G} \delta_i(X_i - \overline{\mu})\|_{\mathcal{K}} \leq O\left(\frac{\epsilon}{\sqrt{k}}\sqrt{\log 1/\epsilon} + \epsilon \cdot \omega\right) \quad (14)$$

where the second step follows by the first part of Condition (I) in the definition of $\epsilon$-goodness for $S_G$, together with the second part of Lemma 3.8.

Next, we bound the first term on the right-hand side of (13). We have

$$|\langle A(w, \overline{\mu}), \Sigma\rangle| \leq |\langle A(\hat{w}(S_G), \overline{\mu}), \Sigma\rangle| + |\langle A(\delta, \overline{\mu}), \Sigma\rangle|$$

$$\leq |\langle A(\hat{w}(S_G), \overline{\mu}), \Sigma\rangle| + O\left(\frac{\epsilon}{k}\log 1/\epsilon + \epsilon \cdot \omega^2\right)$$

$$\leq |\langle B(\{\mu_i\}), \Sigma\rangle| + O\left(\omega^2 + \frac{\epsilon \log 1/\epsilon}{k}\right)$$

$$\leq |\langle B(\hat{\mu}(S_G)), \Sigma\rangle| + O\left(\omega^2 + \frac{\epsilon \log 1/\epsilon}{k}\right), \quad (15)$$

where the second step follows by the third part of Lemma 3.8, the third step follows by Corollary 3.5, and the fourth step follows by Condition (III) of $\epsilon$-goodness.

Additionally, by Corollary 2.5, we can bound

$$|\langle B(\mu(w)), \Sigma\rangle - \langle B(\hat{\mu}(S_G)), \Sigma\rangle| \leq \frac{3}{k}\|\mu(w) - \hat{\mu}(S_G)\|_1 \leq \frac{3}{k}\|w - \hat{w}(S_G)\|_1 \leq O(\epsilon/k). \quad (16)$$

By (15) and (16) we conclude that $\langle A(w, \overline{\mu}), \Sigma\rangle \leq \langle B(\mu(w)), \Sigma\rangle + O(\frac{\epsilon}{k}\log 1/\epsilon)$, so this together with (13) and (14) yields the desired bound. $\qquad\square$

## 3.5 Analyzing the Filter With Spectral Signatures

We now use Lemma 3.7 to show that under the deterministic condition that the uncorrupted points are $\epsilon$-good, LEARNWITHFILTER satisfies the guarantees of Theorem 3.1.

The main step is to show that as long as we remain in the main loop of LEARNWITHFILTER, and we have so far thrown out more bad weight than good weight, we are guaranteed to throw out more bad weight than good weight in the next iteration of the main loop:

**Lemma 3.10.** *Let $w$ and $w'$ be the weights at the start and end of a single iteration of the main loop of* LEARNWITHFILTER. *There is an absolute constant $C > 0$ such that if $\|M(w)\|_{\mathcal{K}} > C \cdot \frac{\epsilon}{k}\log 1/\epsilon$ and $\sum_{i \in S_G} \frac{1}{N} - w_i < \sum_{i \in S_B} \frac{1}{N} - w_i$, then $\sum_{i \in S_G} w_i - w'_i < \sum_{i \in S_B} w_i - w'_i$.*

*Proof.* Suppose the scores $\tau_1, \cdots, \tau_N$ in this iteration are sorted in decreasing order, and let $T$ denote the smallest index for which $\sum_{i \in [T]} w_i \geq 2\epsilon$. As FILTER does not modify $w_i$ for $i > T$, we just need to show that $\sum_{i \in S_G \cap [T]} w_i - w'_i < \sum_{i \in S_B \cap [T]} w_i - w'_i$, and by Lemma 3.2 it is enough to show that

$$\sum_{i \in S_G \cap [T]} w_i \tau_i < \sum_{i \in S_B \cap [T]} w_i \tau_i. \quad (17)$$

First note that because each weight is at most $\epsilon$, we may assume that $\sum_{i \in [T]} w_i \leq 3\epsilon$. We begin by upper bounding the left-hand side of (17).

**Lemma 3.11.** $\sum_{i \in S_G \cap [T]} w_i \tau_i \leq O\left(\frac{\epsilon}{k} \log 1/\epsilon + \epsilon \cdot \omega^2 + \epsilon^2 \|M(w)\|_{\mathcal{K}}\right).$

*Proof.* Let $w''$ be the weights given by $w''_i$ for $i \in S_G \cap [T]$ and $w''_i = 0$ otherwise. Then $\sum_{S_G \cap [T]} w_i \tau_i$ is equal to

$$
\sum_{i \in [N]} w''_i \tau_i = \sum_{i \in [N]} w''_i \left\langle (X_i - \mu(w))^{\otimes 2}, \Sigma \right\rangle
$$

$$
= \sum_{i \in [N]} w''_i \left\langle (X_i - \mu(w''))^{\otimes 2}, \Sigma \right\rangle + \|w''\|_1 \cdot \left\langle (\mu(w'') - \mu(w))^{\otimes 2}, \Sigma \right\rangle \tag{18}
$$

$$
\leq \sum_{i \in [N]} w''_i \left\langle (X_i - \mu(w''))^{\otimes 2}, \Sigma \right\rangle + O(\epsilon) \cdot \|\mu(w'') - \mu(w)\|_{\mathcal{K}}^2 \tag{19}
$$

$$
\leq \sum_{i \in [N]} w''_i \left\langle (X_i - \overline{\mu})^{\otimes 2}, \Sigma \right\rangle + O(\epsilon) \cdot \|\mu(w'') - \mu(w)\|_{\mathcal{K}}^2 \tag{20}
$$

$$
\leq O\left(\epsilon \cdot \omega^2 + \frac{\epsilon}{k} \log 1/\epsilon\right) + O(\epsilon) \cdot \|\mu(w'') - \mu(w)\|_{\mathcal{K}}^2
$$

where (18) and (20) both follow from Fact 1.2, (19) follows from the earlier assumption that $\sum_{i \in [T]} w_i \leq 3\epsilon$ and the definition of $\|\cdot\|_{\mathcal{K}}$, and the last step follows by the third part of Lemma 3.8.

Now note that

$$
\|\mu(w'') - \mu(w)\|_{\mathcal{K}} \leq \|\mu(w'') - \overline{\mu}\|_{\mathcal{K}} + \|\mu(w) - \overline{\mu}\|_{\mathcal{K}}
$$

$$
\leq O\left(\frac{\sqrt{\log 1/\epsilon}}{\sqrt{k}} + \omega\right) + \|\mu(w) - \overline{\mu}\|_{\mathcal{K}}
$$

$$
\leq O\left(\frac{\sqrt{\log 1/\epsilon}}{\sqrt{k}} + \omega + \sqrt{\epsilon\left(\|M(w)\|_{\mathcal{K}} + \omega^2 + \frac{\epsilon}{k} \log 1/\epsilon\right)}\right),
$$

where the second step follows by the fourth part of Lemma 3.8 and the third step holds by Lemma 3.7. The desired bound follows. $\qquad\square$

One consequence of this is that outside of the tails, the scores among good samples are small.

**Corollary 3.12.** *For all $i > T$, $\tau_i \leq O(\frac{1}{k} \log 1/\epsilon + \epsilon \|M(w)\|_{\mathcal{K}} + \omega^2)$.*

*Proof.* Note that

$$
\sum_{i \in S_G \cap [T]} w_i = \sum_{i \in [T]} w_i - \sum_{i \in S_B \cap [T]} w_i \geq 2\epsilon - \sum_{i \in S_B} w_i \geq \epsilon,
$$

so the claim follows from Lemma 3.11 and averaging. $\qquad\square$

Next, we show that the deviation of the total scores of the good points from their expectation is negligible.

**Lemma 3.13.** $\sum_{i \in S_G} w_i \tau_i - \langle B(\mu(w)), \Sigma \rangle \leq O\left(\frac{\epsilon}{k} \log 1/\epsilon + \epsilon \cdot \omega^2 + \epsilon \cdot \|M(w)\|_{\mathcal{K}}\right).$

*Proof.* Let $w'$ be the weights given by $w'_i = w_i$ for $i \in S_G$ and $w'_i = 0$ otherwise. Then by Fact 1.2,

$$
\sum_{i \in S_G} w_i \tau_i = \sum_{i \in S_G} w_i \langle (X_i - \mu(w'))^{\otimes 2}, \Sigma \rangle + \|w\|_1 \cdot \langle (\mu(w) - \mu(w'))^{\otimes 2}, \Sigma \rangle
$$

$$
\leq \frac{1}{\sum_{i \in S_G} w_i} \left(\langle B(\mu(w')), \Sigma \rangle + O\left(\frac{\epsilon}{k} \log 1/\epsilon\right)\right) + \|\mu(w) - \mu(w')\|_{\mathcal{K}}^2
$$

where in the second step we used Fact 1.2, and in the third step we used Lemma 3.9 and the definition of $\|\cdot\|_{\mathcal{K}}$. To bound the $\|\mu(w) - \mu(w')\|_{\mathcal{K}}^2$ term, note that

$$
\begin{aligned}
\|\mu(w) - \mu(w')\|_{\mathcal{K}} &\leq \|\mu(w) - \overline{\mu}\|_{\mathcal{K}} + \|\mu(w') - \overline{\mu}\|_{\mathcal{K}} \\
&\leq \|\mu(w) - \overline{\mu}\|_{\mathcal{K}} + O\left(\frac{\epsilon\sqrt{\log 1/\epsilon}}{\sqrt{k}} + \epsilon \cdot \omega\right) \\
&\leq O\left(\frac{\epsilon\sqrt{\log 1/\epsilon}}{\sqrt{k}} + \epsilon \cdot \omega + \sqrt{\epsilon\left(\|M(w)\|_{\mathcal{K}} + \omega^2 + \frac{\epsilon}{k}\log 1/\epsilon\right)}\right),
\end{aligned}
$$

where the second step follows by the fourth part of Lemma 3.8, and the third step follows by Lemma 3.7. Finally, by Corollary 2.5 we have that

$$
\langle B(\mu(w')), \Sigma\rangle \leq \langle B(\mu(w)), \Sigma\rangle + \frac{3}{k}\|\mu(w') - \mu(w)\|_1 \leq \langle B(\mu(w)), \Sigma\rangle + O(\epsilon/k),
$$

where the last step follows by Fact 1.3. This completes the proof of the claim. $\qquad\square$

We are now ready to complete the proof of Lemma 3.10. In light of Lemma 3.11, we wish to lower bound the right-hand side of (17).

**Claim 3.14.** *If $C > 0$ in the lower bound $\|M(w)\|_{\mathcal{K}} > C(\frac{\epsilon}{k}\log 1/\epsilon + \omega^2)$ is sufficiently large, then $\langle M(w), \Sigma^*\rangle$ must be positive.*

*Proof.* Let $w'$ denote the weights given by $w'_i = w_i$ for $i \in S_G$ and $w'_i = 0$ otherwise. We have

$$
\begin{aligned}
M(w) &= \sum_{i\in[N]} w_i(X_i - \mu(w))^{\otimes 2} - B(\mu(w)) \\
&\succeq \sum_{i\in S_G} w'_i(X_i - \mu(w))^{\otimes 2} - B(\mu(w)) \\
&\succeq \sum_{i\in S_G} w'_i(X_i - \mu(w'))^{\otimes 2} - B(\mu(w)) \\
&= M(w') + B(\mu(w')) - B(\mu(w)) \tag{21}
\end{aligned}
$$

where the third step follows by Fact 1.2. Furthermore,

$$
\|B(\mu(w')) - B(\mu(w))\|_{\mathcal{K}} \leq \frac{3}{k}\cdot\|\mu(w') - \mu(w)\|_1 \leq O(\epsilon/k) \tag{22}
$$

by Corollary 2.5 and Fact 1.3. Lastly, we must bound $\|M(w')\|_{\mathcal{K}}$. Letting $\hat{w}'$ denote the normalized version of $w'$, we have that

$$
\begin{aligned}
\|M(w')\|_{\mathcal{K}} &\leq \|M(\hat{w}')\|_{\mathcal{K}} + \|M(w') - M(\hat{w}')\|_{\mathcal{K}} \\
&\leq \|M(\hat{w}')\|_{\mathcal{K}} + \|A(\hat{w}' - w', \overline{\mu})\|_{\mathcal{K}} \\
&\leq O\left(\frac{\epsilon}{k}\log 1/\epsilon + \omega^2\right), \tag{23}
\end{aligned}
$$

where the penultimate step follows by Fact 1.2 and the definition of the matrix $M(\cdot)$, and the last step follows by Lemma 3.9 and the third part of Lemma 3.8.

We conclude by (21), (22), and (23) that

$$
\min_{\Sigma\in\mathcal{K}}\langle M(w), \Sigma\rangle \geq -O\left(\frac{\epsilon}{k}\log 1/\epsilon + \omega^2\right), \tag{24}
$$

so we simply need to take $C$ larger than the constant implicit in the right-hand side of (24) to ensure that $\langle M(w), \Sigma^*\rangle > 0$. $\qquad\square$

By Claim 3.14 and the definition of the scores,

$$
\sum_{i\in[N]} w_i\tau_i - \langle B(\mu(w)), \Sigma^*\rangle = \langle M(w), \Sigma^*\rangle \geq \|M(w)\|_{\mathcal{K}}.
$$

This, together with Lemma 3.13, yields $\sum_{i \in S_B} w_i \tau_i \geq C' \|M(w)\|_{\mathcal{K}}$ for some $C' < C$ which we can take to be arbitrarily large. We want to show that this same sum, over only $S_B \cap [T]$, enjoys essentially the same bound. Indeed,

$$\sum_{i \in S_B \cap [T]} w_i \tau_i \geq C' \|M(w)\|_{\mathcal{K}} - \sum_{i \in S_B \setminus [T]} w_i \tau_i$$
$$\geq C' \|M(w)\|_{\mathcal{K}} - \left( \sum_{i \in S_B} w_i \right) \cdot O\left( \frac{1}{k} \log 1/\epsilon + \omega^2 + \epsilon \|M(w)\|_{\mathcal{K}} \right)$$
$$\geq \overline{C} \cdot \|M(w)\|_{\mathcal{K}},$$

for some arbitrarily large absolute constant $\overline{C}$, where the second step follows by Corollary 3.12, and the last by the assumption that $\|M(w)\|_{\mathcal{K}} > C \cdot (\frac{\epsilon}{k} \log 1/\epsilon + \omega^2)$. On the other hand, by this same assumption and by Lemma 3.11,

$$\sum_{i \in S_G \cap [T]} w_i \tau_i \leq O\left( \frac{\epsilon}{k} \log 1/\epsilon + \epsilon \cdot \omega^2 + \epsilon^2 \|M(w)\|_{\mathcal{K}} \right) \leq \underline{C} \cdot \|M(w)\|_{\mathcal{K}},$$

where $\underline{C}$ can be taken to be smaller than $\overline{C}$. This proves (17) and thus Lemma 3.10. $\qquad\square$

We can now combine Lemma 3.7 and Lemma 3.10 to get a proof of Theorem 3.1.

*Proof of Theorem 3.1.* Let $\hat{\mu}$ be the output of LEARNWITHFILTER. By Lemma 1.7, it suffices to show that $\hat{\mu}$ satisfies $\|\hat{\mu} - \mu\|_{\mathcal{A}_{s(d+1)}} \leq O(\omega + \frac{\epsilon}{\sqrt{k}} \sqrt{\log 1/\epsilon})$, or equivalently that for all $v \in \mathcal{V}_\ell^n$, where $\ell \triangleq 2s(d+1)$, we have that $\langle (\hat{\mu} - \mu)^{\otimes 2}, vv^\top \rangle^{1/2} \leq O(\omega + \frac{\epsilon}{\sqrt{k}} \sqrt{\log 1/\epsilon})$. By Corollary 2.3, it is enough to show that $\|\hat{\mu} - \mu\|_{\mathcal{K}} \leq O(\omega + \frac{\epsilon}{\sqrt{k}} \sqrt{\log 1/\epsilon})$. By Lemma 3.7 together with the termination condition of the main loop of LEARNWITHFILTER, we just need to show that the algorithm terminates (in polynomial time) and that $w \in \mathcal{W}_{O(\epsilon)}$.

But by induction and Lemma 3.10, every iteration of the loop removes more mass from the bad points than from the good points. Furthermore, by Lemma 3.2, the support of $w$ goes down by at least one every time 1DFILTER is run, so the loops terminates after at most $N$ iterations, each of which can be implemented in polynomial time. At the end, at most an $\epsilon$ fraction of the total mass on $S_G$ has been removed, so the final weights $w$ satisfy $w \in \mathcal{W}_{2\epsilon}$ as desired. $\qquad\square$

# 4 Numerical Experiments

In this section we report on empirical evaluations of our algorithm on synthetic data. We compared our algorithm LEARNWITHFILTER, the naive estimator which simply takes the empirical mean of all samples, the "oracle" algorithm which computes the empirical mean of the *uncorrupted samples*, and the threshold of $\epsilon/\sqrt{k}$ which our theorems show that LEARNWITHFILTER achieves, up to constant factors (in Figures 1 and 2, these are labeled "filter", "naive", "oracle", and $\epsilon/\sqrt{k}$ respectively). Note that by definition, the oracle dominates the algorithms considered in [CLM19] and [JO19] for the *unstructured* case, as those algorithms search for a subset of the data and output the empirical mean of that subset. But as Theorem 3.1 predicts, LEARNWITHFILTER should actually *outperform* the oracle in settings where the underlying distribution $\mu$ is *structured* and there are too few samples for the empirical mean of the uncorrupted points to concentrate sufficiently. In these experiments, we confirm this empirically.

## 4.1 Experimental Design

Our experiments fall under two types: (A) those on learning an *arbitrary distribution* in $\mathcal{A}_{\ell/2}$ *norm* and B) those on learning a *structured distribution* in *total variation distance*. The purpose of experiments of type (A) will be to convey that LEARNWITHFILTER can be used to learn from untrusted batches in $\mathcal{A}_{\ell/2}$ norm even for distributions which are not necessarily structured. The purpose of experiments of type (B) will be to demonstrate that LEARNWITHFILTER can outperform the oracle for structured distributions.

Figure 1: **Arbitrary Distributions**:

Throughout, $\omega = 0$ and $\ell = 10$. While our algorithm can also be implemented for larger $\ell$ (as the size of the SDP we solve does not depend on $\ell$), we choose $\ell = 5$ because it is small enough that the sample complexity savings of our algorithm are very pronounced, yet large enough that for the domain sizes $n$ we work with, enumerating over $\mathcal{V}_\ell^n$ would be prohibitively expensive, justifying the need to use an SDP.

For experiments of type (A), we chose the true underlying distribution $\mu$ by sampling uniformly from $[0,1]^n$ and normalizing, and for experiments of type B), we chose $\mu$ by sampling a uniformly random piecewise constant function with $\ell = 5$ pieces.

Given $\mu$ and a prescribed parameter $\delta$, the distribution from which the corrupted batches were drawn was taken to be $\mathrm{Mul}_k(\nu)$, where $\nu$ was constructed to satisfy $d_{\mathrm{TV}}(\mu, \nu) = \delta$ by adding $\frac{2\delta}{n}$ to the smallest entries of $\mu$ and subtracting $\frac{2\delta}{n}$ from the largest. Sometimes this does not give a probability distribution, in which case we resample $\mu$. When $k, \epsilon, N$ are clear from context and we say that $N$ $\epsilon$-corrupted batches are drawn from the distribution specified by $(\mu, \nu)$, we mean that $\lfloor (1-\epsilon)N \rfloor$ samples are drawn from $\mathrm{Mul}_k(\mu)$ and $N - \lfloor (1-\epsilon)N \rfloor$ from $\mathrm{Mul}_k(\nu)$.

As noted in [JO19], choosing $\delta$ too high makes it too easy to detect the corruptions in the data, while choosing $\delta$ too low means the naive estimator will already perform quite well. In light of this and the fact that the above process for generating $\nu$ only ensures that $d_{\mathrm{TV}}(\mu, \nu) = \delta$, whereas $\|\mu - \nu\|_{\mathcal{A}_\ell}$ might be much smaller, we chose $\delta$ for our experiments as follows. For experiments of type (A), we took $\delta = 0.5$ to ensure that the typical $\mathcal{A}_{\ell/2}$ distance between the empirical mean and the truth was still sufficiently large that the the naive estimator was not competitive. For experiments of type B) where we measure error in terms of total variation distance, we could afford to choose $\delta$ slightly smaller, namely $\delta = 0.3$.

We first describe the experiments of type (A). We examined the effect of varying one of the following four parameters at a time: domain size $n$, batch size $k$, corruption fraction $\epsilon$, and total number of batches $N$. Each of the following four experiments was repeated for a total of ten trials.

Figure 2: **Structured Distributions**:

(a) *Varying domain size* $n$: We fixed $\epsilon = 0.4$, $k = 1000$, and $N = \lfloor \frac{\ell/\epsilon^2}{1-\epsilon} \rfloor$ to ensure $\lfloor \ell/\epsilon^2 \rfloor$ samples from $\text{Mul}_k(\mu)$. We chose such large $k$ to ensure the gap between empirical mean and our algorithm was very noticable. In each trial and for each $n \in [4, 8, 16, 32, 64, 128]$, we randomly generated $(\mu, \nu)$ via the above procedure, drew $N$ $\epsilon$-corrupted samples from distribution specified by $(\mu, \nu)$. Note that while $N$ is independent of $n$, the performance of our algorithm is comparable to that of the oracle.[1]

(b) *Varying batch size* $k$: We fixed $\epsilon = 0.4$, $n = 64$, and $N = \frac{\ell/\epsilon^2}{1-\epsilon} \rfloor$. In each trial, we randomly generated $(\mu, \nu)$ via the above procedure, and then for each value of $k \in [1, 50, 100, 250, 500, 750, 1000]$ we drew $N$ samples from the distribution specified by $(\mu, \nu)$. Note that while our algorithm's error and the oracle's error decay with $k$, the empirical mean's error remains fixed.

(c) *Varying corruption fraction* $\epsilon$: We fixed $\epsilon^* = 0.4$, $n = 64$, $k = 1000$, and $N = \lfloor \ell/\epsilon^{*2} \rfloor$. In each trial, we randomly generated $(\mu, \nu)$ via the above procedure and drew $N$ samples from $\text{Mul}(k, \mu)$. Then for each $\epsilon \in [0.0, 0.1, 0.2, 0.3, 0.4]$, we augmented this with an additional $\lfloor \frac{\epsilon N}{1-\epsilon} \rfloor$ samples from $\text{Mul}(k, \nu)$. Note that while our algorithm's error remains close to $\epsilon^*/\sqrt{k}$, the empirical mean's error increases linearly in $\epsilon$.

(d) *Varying number of batches* $N$: We fixed $\epsilon = 0.4$, $n = 128$, and $k = 500$. In each trial, we randomly generated $(\mu, \nu)$ via the above procedure, and then for each $\rho \in [0.5, 0.75, 1, 1.25, 1.5]$, we drew $N = \lfloor \rho \cdot \ell/\epsilon^2 \rfloor$ samples from the distribution specified by $(\mu, \nu)$. Note that even with such a small number of samples, our algorithm can compete with the oracle. Also note that our error bottoms out at $\epsilon/\sqrt{k}$ while the oracle's error goes beneath this threshold.

For type (B), we ran the exact same set of four experiments but over structured $\mu$, with the key difference that after generating an estimate with LEARNWITHFILTER, we post-processed it by

rounding to a piecewise constant function via a simple dynamic program. We then compare the error of this piecewise constant estimator in *total variation distance* to that of the empirical mean of the whole dataset, and the empirical mean of the uncorrupted points.

As is evident from Figure 2, our algorithm outperforms even the oracle, as predicted by Theorem 3.1.

## 4.2 Implementation Details

The experiments were conducted on a MacBook Pro with 2.6 GHz Dual-Core Intel Core i5 processor and 8 GB of RAM. The experiments of type (A) respectively took 110m36.499s, 73m19.477s, 50m54.655s, and 536m39.212s to run. The experiments of type (B) respectively took 64m28.346s, 52m7.859s, 39m36.754s, and 362m50.742s to run. The discrepancy in runtimes between (A) and (B) can be explained by the fact that a number of unrelated processes were also running at the time of the former. The experiment of varying the number of batches $N$ was the most expensive because we chose domain size $n = 128$ to accentuate the gap between our algorithm and the oracle. The abovementioned runtimes imply that over a domain of size 128, LEARNWITHFILTER takes roughly 7-10 minutes.

For the implementation, we used the SCS solver in CVXPY for our semidefinite programs. In order to achieve reasonable runtimes, we needed to set the feasibility tolerance to $1e - 2$, and as a result the SDP solver would occasionally output matrices $\Sigma$ which are moderately far from $\mathcal{K}$; in particular, one mode of failure that arose was that $\Sigma$ might be non-PSD and give rise to negative scores in LEARNWITHFILTER. We chose to address this mode of failure heuristically by terminating the algorithm whenever this happened and simply outputting the estimate for $\mu$ at that point in time. Of the 480 total trials that were run across all experiments, this happened 53 times. Another heuristic that we used was to terminate the algorithm as soon as $\|\Sigma\|_{\mathcal{K}}$ stopped increasing during a run of LEARNWITHFILTER; this was primarily to have a stopping criterion that avoids the need to tune constant factors. As demonstrated by Figures 1 and 2, these heuristic decisions ultimately had negligible effect on the performance of our algorithm.

## Footnotes

[1]The naive estimator's error is decreasing in $n$ for an unrelated reason: as $n$ increases, the above procedure for sampling $(\mu, \nu)$ appears to skew towards $\mu$ for which the resulting perturbation $\nu$ is close in $\mathcal{A}_{\ell/2}$.

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

## A  Concentration

In this section we prove Lemma 3.6, restated here for convenience:

**Lemma 3.6** (Regularity of good samples). *If $U$ is a set of $\widetilde{\Omega}\left(\log(1/\delta)(\ell^2/\epsilon^2) \cdot \log^3(n)\right)$ independent samples from $Mul_k(\mu_1), ..., Mul_k(\mu_{|U|})$, then $U$ is $\epsilon$-good with probability at least $1 - \delta$.*

## A.1 Technical Ingredients

The key technical fact we use to get sample complexity that depend quadratically on $\ell$ is:

**Lemma A.1.** *For every $0 < \eta \le 1$, there exists a net $\mathcal{N} \subset \mathbb{R}^{n \times n}$ of size $O(n^3 \ell^2 \log^2 n / \eta)^{(\ell \log n + 1)^2}$ of matrices such that for every $\Sigma \in \mathcal{K}$, there exists some $\tilde{\Sigma} = \sum_\nu \Sigma_\nu^*$ for $\Sigma_\nu^* \in \mathcal{N}$ such that the following holds: 1) $\|\Sigma - \tilde{\Sigma}\|_F \le \eta$, 2) $\sum_\nu \alpha_\nu \le 1$, and 3) $\|\Sigma_\nu^*\|_{\max} \le O(1)$.*

Note that this is a strengthening of a special case of Lemma 6.9 from [CLM19]. We defer the proof of Lemma A.1 to Appendix B.

For $\epsilon$-goodness to hold, it will be crucial to establish the following sub-exponential tail bounds for the empirical covariance of a set of samples $X_1, \cdots, X_N \sim \mathrm{Mul}_k(\mu)$, as well as for $\|\hat{\mu} - \mu\|_{\mathcal{K}}^2$, where $\hat{\mu}$ is the empirical mean of those samples.

**Lemma A.2.** *Let $\xi > 0$ and let $\mathcal{N} \subset \mathbb{R}^{n \times n}$ be any finite set for which $\|\Sigma\|_{\max} \le O(1)$ for all $\Sigma \in \mathcal{N}$. Let $\mu_1, ..., \mu_N, \overline{\mu} \in \Delta^n$ satisfy $\overline{\mu} \triangleq \frac{1}{N} \sum_{i=1}^N \mu_i$. Then for $X_i \sim \mathrm{Mul}_k(\mu_i)$ for $i \in [N]$,*

$$
\Pr \left[ \left| \left\langle \frac{1}{N} \sum_{i=1}^N (X_i - \mu_i)^{\otimes 2} - \mathop{\mathbb{E}}_{X \sim \mathrm{Mul}_k(\mu_i)} \left[ (X - \mu_i)^{\otimes 2} \right], \Sigma \right\rangle \right| > t \; \forall \Sigma \in \mathcal{N} \right] < 2|\mathcal{N}| \exp \left( -\Omega \left( \frac{N k^2 t^2}{1 + kt} \right) \right),
$$

*where the probability is over the samples $X_1, \cdots, X_N$.*

**Lemma A.3.** *Let $\xi > 0$ and let $\mathcal{N} \subset \mathbb{R}^{n \times n}$ be any finite set for which $\|\Sigma\|_{\max} \le O(1)$ for all $\Sigma \in \mathcal{N}$. For $X_i \sim \mathrm{Mul}_k(\mu_i)$ for $i \in [N]$, $\hat{\mu} \triangleq \frac{1}{N} \sum_{i=1}^N X_i$, and $\overline{\mu} \triangleq \frac{1}{N} \sum_{i=1}^N \mu_i$,*

$$
\Pr \left[ \left| \langle (\hat{\mu} - \mu)^{\otimes 2}, \Sigma \rangle - \mathbb{E} \left[ \langle (\hat{\mu} - \mu)^{\otimes 2}, \Sigma \rangle \right] \right| > t \; \forall \Sigma \in \mathcal{N} \right] < 2|\mathcal{N}| \exp \left( -\Omega \left( \frac{N^2 k^2 t^2}{1 + Nkt} \right) \right),
$$

*where the probability is over the samples $X_1, \cdots, X_N$.*

**Lemma A.4.** *Let $\xi > 0$ and let $\mathcal{N} \subset \mathbb{R}^{n \times n}$ be any finite set for which $\|\Sigma\|_{\max} \le O(1)$ for all $\Sigma \in \mathcal{N}$. Let $\mu_1, ..., \mu_N, \overline{\mu} \in \Delta^n$ satisfy $\|\mu_i - \overline{\mu}\|_1 \le \omega$ for all $i \in [N]$. For $X_i \sim \mathrm{Mul}_k(\mu_i)$ for $i \in [N]$,*

$$
\Pr \left[ \left| \frac{1}{N} \sum_{i=1}^N (\mu_i - \overline{\mu})^\top \Sigma (X_i - \mu_i) \right| > \omega \cdot t \; \forall \Sigma \in \mathcal{N} \right] < 2|\mathcal{N}| \exp \left( -\Omega \left( kNt^2 \right) \right),
$$

*where the probability is over the samples $X_1, \cdots, X_N$.*

Note that if $\mathcal{N}$ consisted solely of matrices of the form $vv^\top$ for $v \in \{\pm 1\}^n$, these lemmas would follow straightforwardly from standard binomial tail bounds. Instead, we only have entrywise bounds for the matrices in $\mathcal{N}$ and will therefore need to compute moment estimates from scratch in order to prove Lemmas A.2 and A.3. We defer the details of this to Appendix C.

Lastly, we will need the following elementary consequence of Stirling's formula:

**Fact A.5.** *For any $m \ge 1$, $\log \binom{m}{\epsilon m} \le 2m \cdot \epsilon \log 1/\epsilon$.*

## A.2 Proof of Lemma 3.6

We are now ready to prove that the four conditions for $\epsilon$-goodness hold for a set $U$ of independent draws from $\mathrm{Mul}_k(\mu_1), ..., \mathrm{Mul}_k(\mu_{|U|})$ respectively, of size

$$
|U| = \widetilde{\Omega} \left( \log(1/\delta)(\ell^2/\epsilon^2) \cdot \log^3(n) \right). \tag{25}
$$

*Proof of Lemma 3.6.* As $\| \cdot \|_{\mathcal{K}}$ is defined as a supremum over $\mathcal{K}$, we will reduce controlling the infinitely many directions in $\mathcal{K}$ to controlling a finite net of such directions by invoking Lemma A.1. Specifically, recall that for any $\Sigma \in \mathcal{K}$, by Lemma A.1, there is some $\tilde{\Sigma} = \sum_\nu \alpha_\nu \Sigma_\nu^*$ such that $\Sigma_\nu^* \in \mathcal{N}$ and $\|\Sigma - \tilde{\Sigma}\|_F \le \eta$.

**(Condition (I))** By Lemma A.3, with probability at least $1 - 2|\mathcal{N}| \exp\left(-\Omega(\frac{N^2 k^2 t^2}{1 + Nkt})\right)$, we have that for all $\Sigma \in \mathcal{K}$,

$$
\begin{aligned}
\left\langle (\mu(U) - \overline{\mu})^{\otimes 2}, \Sigma \right\rangle &\leq \left\langle (\mu(U) - \overline{\mu})^{\otimes 2}, \tilde{\Sigma} \right\rangle + \|\mu(U) - \overline{\mu}\|_2^2 \cdot \|\Sigma - \tilde{\Sigma}\|_F \\
&\leq \left\langle (\mu(U) - \overline{\mu})^{\otimes 2}, \tilde{\Sigma} \right\rangle + 2\eta \\
&= \sum_\nu \alpha_\nu \left\langle (\mu(U) - \overline{\mu})^{\otimes 2}, \Sigma_\nu^* \right\rangle + 2\eta \\
&\leq \frac{1}{N} \sum_{i=1}^N \mathbb{E}\left[ \langle (X - \mu_i)^{\otimes 2}, \Sigma_\nu^* \rangle \right] + \sum_\nu \alpha_\nu \cdot t + 2\eta \\
&\leq O(1/k|U|) + t + 2\eta, \tag{26}
\end{aligned}
$$

where the first step follows by Cauchy-Schwarz and triangle inequality, the second step follows by the trivial bound $\|\mu(U) - \mu_i\|_2^2 \leq 2$ and the bound on $\|\Sigma - \tilde{\Sigma}\|_F$ guaranteed by Lemma A.1, the fourth step holds with the claimed probability by Lemma A.3 and the fact that $\|\Sigma_\nu^*\|_{\max} \leq O(1)$ for all $\nu$ by the guarantees of Lemma A.1, and the last step follows by the bound on $\sum \alpha_\nu$ by the guarantees of Lemma A.1, as well as the moment bound in Lemma C.2 applied to $r = 1$.

If $|U|$ satisfies (25) and $\eta, t = O(\frac{\epsilon^2}{k} \log 1/\epsilon)$, the first part of Condition (I) holds.

For the second part, by the steps leading to (26), a union bound over the $\binom{|U|}{\epsilon|U|}$ subsets $W$ and Fact A.5, with probability at least

$$
1 - 2\exp(2|U| \cdot \epsilon \log 1/\epsilon) \cdot |\mathcal{N}| \exp\left(-\Omega\left(\frac{\epsilon^2 |U|^2 k^2 t^2}{1 + \epsilon|U|kt}\right)\right)
$$

we have that $\|\mu(W) - \overline{\mu}_W\|_{\mathcal{K}}^2 \leq O\left(\frac{1}{\epsilon k |U|}\right) + t + 2\eta$ for all $W$. Note that $2 \log 1/\epsilon \leq O\left(\frac{\epsilon|U|^2 k^2 t^2}{1 + \epsilon|U|kt}\right)$ provided $t = \Omega\left(\frac{\log 1/\epsilon}{k}\right)$, so if $|U|$ satisfies (25) and $\eta = O(\frac{\log 1/\epsilon}{k})$, the second part of Condition (I) holds.

**(Condition (II))** For the first part, let $\hat{\mathbf{M}} \triangleq M(\hat{w}(U), \{\mu_i\}_{i \in U})$. By Lemma A.2, with probability at least $1 - 2|\mathcal{N}| \exp\left(-\Omega\left(\frac{|U|k^2 t^2}{1 + kt}\right)\right)$, we have that for all $\Sigma \in \mathcal{K}$,

$$
\begin{aligned}
\langle \hat{\mathbf{M}}, \Sigma \rangle &\leq \langle \hat{\mathbf{M}}, \tilde{\Sigma} \rangle + \|\hat{\mathbf{M}}\|_F \cdot \|\Sigma - \tilde{\Sigma}\|_F \\
&\leq \langle \hat{\mathbf{M}}, \tilde{\Sigma} \rangle + 3\eta \\
&\leq \sum_\nu \alpha_\nu \langle \hat{\mathbf{M}}, \Sigma_\nu^* \rangle + 3\eta \\
&\leq \sum_\nu \alpha_\nu \cdot t + 3\eta \\
&\leq t + 3\eta \tag{27}
\end{aligned}
$$

where the first step follows by Cauchy-Schwarz and triangle inequality, and the second step follows by Lemma 1.4 and the bound on $\|\Sigma - \tilde{\Sigma}\|_F$ guaranteed by Lemma A.1, the fourth step holds with the claimed probability by Lemma A.2 and the fact that $\|\Sigma_\nu^*\|_{\max} \leq O(1)$ for all $\nu$ by the guarantees of Lemma A.2, and the last step follows by the bound on $\sum \alpha_\nu$ by the guarantees of Lemma A.1.

If $|U|$ satisfies (25), $\eta = O\left(\frac{\epsilon}{k} \log 1/\epsilon\right)$, $t = O\left(\frac{\epsilon}{k} \log 1/\epsilon\right)$, the first part of Condition (II) holds.

For the second part, first note that it is slightly different from the first part because we do not subtract out $B(\overline{\mu})$, the reason being that $\|B(\overline{\mu})\|_{\mathcal{K}} \leq O(1/k) = o(\frac{\log 1/\epsilon}{k})$, so this term is negligible. By the steps leading to (27), a union bound over the $\binom{|U|}{\epsilon|U|}$ subsets $W$, and Fact A.5, with probability at least

$$
1 - 2|\mathcal{N}| \exp(2\epsilon|U| \log 1/\epsilon) \cdot \exp\left(-\Omega\left(\frac{\epsilon|U|k^2 t^2}{1 + kt}\right)\right),
$$

we have that $\|M(\hat{w}(W), \{\mu_i\}_{i \in W})\|_{\mathcal{K}} \le t + 3\eta$ for all $W$. Note that $2 \log 1/\epsilon \le O\left(\frac{k^2 t^2}{1 + kt}\right)$ provided $t = \Omega\left(\frac{\log 1/\epsilon}{k}\right)$, so if $|U|$ satisfies (25) and $\eta = O\left(\frac{\log 1/\epsilon}{k}\right)$, the second part of Condition (II) holds.

(**Condition (III)**) First note that

$$B(\{\mu_i\}) - B(\overline{\mu}) = \frac{1}{|U|} \sum_{i \in U} \frac{1}{k} \left(\mathrm{diag}(\mu_i - \overline{\mu}) - (\mu_i^{\otimes 2} - \overline{\mu}^{\otimes 2})\right) = -\frac{1}{|U|} \sum_{i \in U} \frac{1}{k} (\mu_i^{\otimes 2} - \overline{\mu}^{\otimes 2}).$$

Also note that

$$\left\langle \Sigma, \frac{1}{|U|} \sum_{i \in U} (\mu_i^{\otimes 2} - \overline{\mu}^{\otimes 2}) \right\rangle = \frac{1}{|U|} \sum_{i \in U} \left\langle (\mu_i - \overline{\mu})^{\otimes 2}, \Sigma \right\rangle \le \max_i \|\mu_i - \overline{\mu}\|_1^2 \le \omega^2,$$

where in the last step we used Fact 2.4. So $\|B(\{\mu_i\}) - B(\overline{\mu})\|_{\mathcal{K}} \le \omega^2/k$.

It remains to bound $\|B(\hat{\mu}(U)) - B(\overline{\mu})\|_{\mathcal{K}}$. As we only need to show extremely mild concentration here, we will not make an effort to obtain tight bounds. Note that by (1),

$$|\langle \Sigma, B(\hat{\mu}(U)) - B(\overline{\mu}) \rangle| \le \frac{1}{k} |\langle \mathrm{diag}(\hat{\mu}(U) - \overline{\mu}), \Sigma \rangle| + \frac{1}{k} \left| \langle \hat{\mu}(U)^{\otimes 2} - \overline{\mu}^{\otimes 2}, \Sigma \rangle \right|. \qquad (28)$$

We have

$$\langle \mathrm{diag}(\hat{\mu}(U) - \overline{\mu}), \Sigma \rangle \le \sum_{\nu} \alpha_{\nu} \langle \mathrm{diag}(\hat{\mu}(U) - \overline{\mu}), \Sigma_{\nu}^* \rangle + \|\Sigma - \tilde{\Sigma}\|_F \cdot \|\hat{\mu}(U) - \overline{\mu}\|_2$$

$$\le \sum_{\nu} \alpha_{\nu} \langle \hat{\mu}(U) - \overline{\mu}, \mathrm{diag}(\Sigma_{\nu}^*) \rangle + O(\eta). \qquad (29)$$

Note that for any $\nu$, $\langle \hat{\mu}(U) - \overline{\mu}, \mathrm{diag}(\Sigma_{\nu}^*) \rangle = \frac{1}{|U|} \sum_{i \in U} Z_i^{\nu}$ for $Z_i^{\nu} \triangleq \langle X_i - \mu_i, \mathrm{diag}(\Sigma_{\nu}^*) \rangle$. These are independent, mean-zero, $O(1)$-bounded random variables, so by Hoeffding's, for any fixed $\nu$ we have that $|\langle \hat{\mu}(U) - \overline{\mu}, \mathrm{diag}(\Sigma_{\nu}^*) \rangle| \le t$ with probability at least $1 - 2\exp(-\Omega(|U|t^2))$. If we union bound over $\mathcal{N}$, then by taking $\eta, t = O(\epsilon)$, and $|U|$ satisfying (25), (29) will be at most $O(\epsilon)$.

We also have that

$$\left| \langle \hat{\mu}(U)^{\otimes 2} - \overline{\mu}^{\otimes 2}, \Sigma \rangle \right| = \left| \langle (\hat{\mu}(U) - \overline{\mu})^{\otimes 2}, \Sigma \rangle - 2\overline{\mu}^{\top} \Sigma (\hat{\mu}(U) - \overline{\mu}) \right|$$

$$\le O\left(\frac{\epsilon^2 \log 1/\epsilon}{k}\right) + 2 \left| \overline{\mu}^{\top} \Sigma (\hat{\mu}(U) - \overline{\mu}) \right|, \qquad (30)$$

where the second step follows by the first part of this lemma. For the other term, we have

$$\overline{\mu}^{\top} \Sigma (\hat{\mu}(U) - \overline{\mu}) \le \sum_{\nu} \alpha_{\nu} \overline{\mu}^{\top} \Sigma_{\nu}^* (\hat{\mu}(U) - \overline{\mu}) + \|\Sigma - \tilde{\Sigma}\|_F \cdot \|\overline{\mu}\|_2 \cdot \|\hat{\mu}(U) - \overline{\mu}\|_2$$

$$\le \sum_{\nu} \alpha_{\nu} \overline{\mu}^{\top} \Sigma_{\nu}^* (\hat{\mu}(U) - \overline{\mu}) + O(\eta). \qquad (31)$$

For any $\nu$, $\overline{\mu}^{\top} \Sigma_{\nu}^* (\hat{\mu}(U) - \overline{\mu}) = \frac{1}{|U|} \sum_{i \in U} W_i^{\nu}$ for $W_i^{\nu} \triangleq \overline{\mu}^{\top} \Sigma_{\nu}^* (X_i - \mu_i)$. These are independent, mean-zero, $O(1)$-bounded random variables, so by Hoeffding's, for any fixed $\nu$, we have that $|\overline{\mu}^{\top} \Sigma (\hat{\mu}(U) - \overline{\mu})| \le t$ with probability at least $1 - 2\exp(-\Omega(|U|t^2))$. If we union bound over $\mathcal{N}$, then by taking $\eta, t = O(\epsilon)$ and $|U|$ satisfying (25) again, (31) and thus (30) will be at most $O(\epsilon)$.

By (28), we thus conclude that $\|B(\hat{\mu}(U)) - B(\overline{\mu})\|_{\mathcal{K}} \le O(\epsilon/k)$ as claimed.

546 (**Condition (IV)**) By Lemma A.4, with probability at least $1 - 2|\mathcal{N}|\exp\left(-\Omega\left(k|U|t^2\right)\right)$, we have
547 that for all $\Sigma \in \mathcal{K}$,

$$
\frac{1}{|U|}\sum_{i \in U}(\mu_i - \overline{\mu})^\top \Sigma(X_i - \mu_i)
$$

$$
\leq \frac{1}{|U|}\sum_{i \in U}(\mu_i - \overline{\mu})^\top \tilde{\Sigma}(X_i - \mu_i) + \frac{1}{|U|}\sum_{i \in U}\|\Sigma - \tilde{\Sigma}\|_F \cdot \|\mu_i - \overline{\mu}\|_2 \cdot \|X_i - \mu_i\|_2
$$

$$
\leq \sum_\nu \alpha_\nu \cdot \frac{1}{|U|}\sum_{i \in U}(\mu_i - \overline{\mu})^\top \Sigma_\nu^*(X_i - \mu_i) + 2\omega \cdot \eta
$$

$$
\leq \sum_\nu \alpha_\nu \cdot t + 2\omega \cdot \eta
$$

$$
\leq \omega \cdot t + 2\omega \cdot \eta \tag{32}
$$

548 where the first step follows by triangle inequality and Cauchy-Schwarz, the second step follows by
549 the bound on $\|\Sigma - \tilde{\Sigma}\|_F$ guaranteed by Lemma A.1 and the assumption that $\|\mu_i - \overline{\mu}\|_2 \leq \omega$, and the
550 third step holds with the claimed probability by Lemma A.4 and the fact that $\|\Sigma_\nu^*\|_{\max} \leq O(1)$ for all
551 $\nu$ by Lemma A.1, and the last step follows by the bound on $\sum \alpha_\nu$ by the guarantees of Lemma A.1.

552 If $|U|$ satisfies (25) and $\eta, t = O\left(\frac{\epsilon\sqrt{\log 1/\epsilon}}{\sqrt{k}}\right)$, the first part of Condition (IV) holds.

553 For the second part, by the steps leading to (32), a union bound over $W$, and Fact A.5, with probability
554 at least

$$
1 - 2|\mathcal{N}|\exp(2\epsilon|U|\log 1/\epsilon) \cdot \exp\left(-\Omega\left(\epsilon k|U|t^2\right)\right),
$$

555 we have that $\frac{1}{|W|}\sum_{i \in W}(\mu_i - \overline{\mu})^\top \Sigma(X_i - \mu_i) \leq \omega \cdot t + 2\omega \cdot \eta$ for all $W$.

556 Note that $2\log 1/\epsilon \leq O(kt^2)$ provided $t = \Omega\left(\frac{\sqrt{\log 1/\epsilon}}{\sqrt{k}}\right)$, so if $|U|$ satisfies (25) and $\eta =$
557 $O\left(\frac{\sqrt{\log 1/\epsilon}}{\sqrt{k}}\right)$, the second part of Condition (IV) holds. $\qquad\square$

## B Netting Over $\mathcal{K}$

559 In this section we prove Lemma A.1, restated here for convenience:

560 **Lemma A.1.** *For every* $0 < \eta \leq 1$*, there exists a net* $\mathcal{N} \subset \mathbb{R}^{n \times n}$ *of size* $O(n^3 \ell^2 \log^2 n/\eta)^{(\ell \log n + 1)^2}$
561 *of matrices such that for every* $\Sigma \in \mathcal{K}$*, there exists some* $\tilde{\Sigma} = \sum_\nu \Sigma_\nu^*$ *for* $\Sigma_\nu^* \in \mathcal{N}$ *such that the*
562 *following holds: 1)* $\|\Sigma - \tilde{\Sigma}\|_F \leq \eta$*, 2)* $\sum_\nu \alpha_\nu \leq 1$*, and 3)* $\|\Sigma_\nu^*\|_{\max} \leq O(1)$*.*

563 As alluded to in Remark 2.2 and Appendix A, we will use the extra Constraints 3 and 4 in the
564 definition of $\mathcal{K}$ to tighten the proof of Lemma 6.9 from [CLM19] to obtain Lemma A.1 above.

565 The following well-known trick will be useful.

566 **Lemma B.1** ("Shelling"). *If* $v \in \mathbb{R}^m$ *satisfies* $\|v\|_2 \leq C$ *and* $\|v\|_1 = C \cdot \sqrt{k}$*, then there exist* $k$*-sparse*
567 *vectors* $v[1], ..., v[m/k]$ *with disjoint supports for which 1)* $v = \sum_{i=1}^{m/k} v[i]$*, 2)* $\sum_{i=1}^{m/k}\|v[i]\|_2 \leq 2C$*,*
568 *and 3)* $\sum_{i=1}^{m/k}\|v[i]\|_\infty \leq \frac{1}{k}\|v\|_1 + \|v\|_\infty$*.*

569 *Proof.* Assume without loss of generality that $C = 1$. Letting $B_1 \subset [m]$ be the indices of the $k$ largest
570 entries of $v$ in absolute value, $B_2$ those of the next $k$ largest, etc., we can write $[m] = B_1 \sqcup \cdots \sqcup B_{m/k}$.
571 For $i \in [m/k]$, define $v[i] \in \mathbb{R}^m$ to be the restriction of $v$ to the coordinates indexed by $B_i$. For any $i$
572 and $j \in B_i$, $|v_j| \leq \frac{1}{k}\|v[i-1]\|_1$. This immediately implies that

$$
\sum_{i=1}^{m/k}\|v[i]\|_\infty \leq \|v\|_\infty + \frac{1}{k}\sum_{i=1}^{m/k}\|v[i]\|_1,
$$

573 yielding 3) above. Likewise, it implies that

$$
\|v[i]\|_2^2 = \sum_{j \in B_i} v_j^2 \leq k \cdot \frac{1}{k^2} \cdot \|v[i-1]\|_1^2 = \frac{1}{k}\|v[i-1]\|_1^2.
$$

574 So $\|v[i]\|_2 \le \|v[i-1]\|_1/\sqrt{k}$ and thus

$$\sum_{i=1}^{m/k} \|v[i]\|_2 \le \|v[1]\|_2 + \frac{1}{\sqrt{k}}\|v\|_1 \le 2,$$

575 giving 2) above. $\qquad\square$

576 By rescaling the entries of $v$ in Lemma B.1, we immediately get the following extension to Haar-
577 weighted norms:

578 **Corollary B.2.** *If $v \in \mathbb{R}^m$ satisfies $\|v\|_{2;\mathbf{h}} \le C$ and $\|v\|_{1;\mathbf{h}} = C \cdot \sqrt{k}$, then there exist $k$-sparse*
579 *vectors $v_1, ..., v_{m/k}$ with disjoint supports for which 1) $v = \sum_{i=1}^{m/k} v_i$, 2) $\sum_{i=1}^{m/k} \|v_i\|_{2;\mathbf{h}} \le 2C$, and*
580 *3) $\sum_{i=1}^{m/k} \|v[i]\|_{\infty;\mathbf{h}} \le \frac{1}{k}\|v\|_{1;\mathbf{h}} + \|v\|_{\infty;\mathbf{h}}$.*

581 We remark that whereas in [CLM19], shelling was applied to the unweighted $L_1$, $L_2$ norms, and the
582 only $L_2$ information used about $v \in \mathcal{V}_\ell^n$ was that $\|v\|_2^2 = n$, in the sequel we will shell under the
583 Haar-weighted norms and use the refined bounds on the Haar-weighted norms given by Constraints 3
584 and 4 from Definition 2.1. This will be crucial to getting a net of size exponential in $\ell^2$ rather than
585 just $\text{poly}(\ell)$.

586 We now complete the proof of Lemma A.1.

587 *Proof of Lemma A.1.* Let $s = \ell \log n + 1$, and let $m = \log n$. Let $\mathcal{N}'$ be an $O\left(\frac{\eta}{n \cdot s^2}\right)$-net in Frobenius
588 norm for all $s^2$-sparse $n \times n$ matrices of unit Frobenius norm. Because $\mathbb{S}^{s^2-1}$ has an $O\left(\frac{\eta}{n \cdot s^2}\right)$-net in
589 $L_2$ norm of size $O(n \cdot s^2/\eta)^{s^2}$, by a union bound we have that

$$|\mathcal{N}'| \le \binom{n^2}{s^2} \cdot O(n \cdot s^2/\eta)^{s^2} = O(n^3 \ell^2 \log^2 n/\eta)^{s^2}$$

590 Take any $\Sigma \in \mathcal{K}$ and consider $\mathbf{L} \triangleq H\Sigma H^\top$. By Constraints 2, 3, 4 in Definition 2.1,

$$\|\mathbf{L}\|_{1,1;\mathbf{h}} \le s^2, \qquad \|\mathbf{L}\|_{F;\mathbf{h}}^2 \le s^2, \qquad \text{and} \qquad \|\mathbf{L}\|_{\max;\mathbf{h}} \le 1. \tag{33}$$

591 We can use the first two of these and apply Corollary B.2 to the $n^2$-dimensional vector $\mathbf{L}$ to conclude
592 that $\mathbf{L} = \sum_j \mathbf{L}^j$ for some matrices $\{\mathbf{L}^j\}_j$ of sparsity at most $s^2$ and for which $\sum_j \|\mathbf{L}^j\|_{F;\mathbf{h}} \le 2s^2$
593 and $\sum_j \|\mathbf{L}^j\|_{\max;\mathbf{h}} \le \frac{1}{s^2}\|\mathbf{L}^j\|_{1,1;\mathbf{h}} + \|\mathbf{L}^j\|_{\max;\mathbf{h}}$.

594 By definition of the Haar-weighted Frobenius norm, $\|\mathbf{L}^j\|_F \le n \cdot \|\mathbf{L}^j\|_{F,\mu}$, so

$$\sum_j \|\mathbf{L}^j\|_F \le O(n \cdot s^2).$$

595 For each $\mathbf{L}^j$, there is some $(\mathbf{L}')^j \in \mathcal{N}'$ such that for $\tilde{\mathbf{L}}^j \triangleq \|\mathbf{L}^j\|_F \cdot (\mathbf{L}')^j$,

$$\|\mathbf{L}^j - \tilde{\mathbf{L}}^j\|_F \le O\left(\frac{\eta}{n \cdot s^2}\right)\|\mathbf{L}^j\|_F. \tag{34}$$

596 We conclude that if we define $\tilde{\mathbf{L}} \triangleq \sum_j \tilde{\mathbf{L}}^j$, then $\|\mathbf{L} - \tilde{\mathbf{L}}\|_F \le \eta$.

597 Now let $\mathcal{N} \triangleq H^{-1}\mathcal{N}^{-1}(H^{-1})^\top$. As $\Sigma = H^{-1}\mathbf{L}(H^{-1})^\top$ and $H^{-1}$ is an isometry, if we define
598 $\tilde{\Sigma}^j \triangleq H^{-1}\tilde{\mathbf{L}}^j(H^{-1})^\top$ and $\tilde{\Sigma} \triangleq \sum_j \tilde{\Sigma}^j$, then we likewise get that $\|\Sigma - \tilde{\Sigma}\|_F \le \eta$, and clearly
599 $\tilde{\Sigma}^j \in \mathbb{P}\mathcal{N}$ for every $j$, concluding the proof of part 1) of the lemma.

600 For each $\tilde{\Sigma}^j$, define
$$\alpha_j \triangleq \|\mathbf{L}^j\|_{\max;\mathbf{h}}/2 \tag{35}$$
601 and define $\Sigma_*^j \triangleq \tilde{\Sigma}^j/\alpha_j$ so that $\tilde{\Sigma} = \sum_{j,\sigma,\tau} \alpha_j \cdot \Sigma_*^j$. Note that by part 3) of Corollary B.2 and (33),

$$\sum_j \alpha_j = \frac{1}{2}\sum_j \|\mathbf{L}^j\|_{\max;\mathbf{h}}$$

$$\le \frac{1}{2}\frac{1}{s^2}\|\mathbf{L}\|_{1,1;\mathbf{h}} + \|\mathbf{L}\|_{\max;\mathbf{h}} \le 1$$

602 where in the last step we used the fact that $\|\mathbf{L}\|_{1,1;\mathbf{h}} \leq s^2$ and $\|\mathbf{L}[\sigma,\tau]\|_{\max;\mathbf{h}} \leq 1$. This concludes
603 the proof of part 2) of the lemma.

604 Finally, we need to bound $\|\Sigma_*^j\|_{\max}$. Note first that for any matrix $\mathbf{J}$ supported only on a submatrix
605 consisting of entries of $\mathbf{L}$ from the rows $i$ (resp. columns $j$) for which $i \in T_\sigma$ (resp. $j \in T_\tau$), we have
606 that

$$\|H^{-1}\mathbf{J}(H^{-1})^\top\|_{\max} = 2^{-(m-\sigma)/2} \cdot 2^{-(m-\tau)/2} \cdot \|\mathbf{J}\|_{\max} = \frac{2^{(\sigma+\tau)/2}}{n}\|\mathbf{J}\|_{\max}$$

607 because the Haar wavelets $\{\psi_{\sigma,j}\}_j$ (resp. $\{\psi_{\tau,j}\}_j$) have disjoint supports and $L_\infty$ norm $2^{-(m-\sigma)/2}$
608 (resp. $2^{-(m-\tau)/2}$). For general $\mathbf{J}$, by decomposing $\mathbf{J}$ into such submatrices, call them $\mathbf{J}[\sigma,\tau]$, we get
609 by triangle inequality that

$$\|H^{-1}\mathbf{J}(H^{-1})^\top\|_{\max} \leq \sum_{\sigma,\tau} \frac{2^{(\sigma+\tau)/2}}{n}\|\mathbf{J}[\sigma,\tau]\|_{\max} \leq \|\mathbf{J}\|_{\max}. \tag{36}$$

610 By applying this to $\mathbf{J} = \tilde{\Sigma}^j$, we get

$$\begin{aligned}
\|\tilde{\Sigma}^j\|_{\max} &\leq \left(\|H^{-1}\mathbf{L}^j(H^{-1})^\top\|_{\max} + \|H^{-1}\left(\mathbf{L}^j - \tilde{\mathbf{L}}^j\right)(H^{-1})^\top\|_{\max}\right)\\
&\leq \|\mathbf{L}^j\|_{\max} + \|\mathbf{L}^j - \tilde{\mathbf{L}}^j\|_{\max}\\
&\leq \|\mathbf{L}^j\|_{\max} + \|\mathbf{L}^j - \tilde{\mathbf{L}}^j\|_F\\
&\leq \|\mathbf{L}^j\|_{\max} + O\left(\frac{\eta}{n\cdot s^2}\right)\|\mathbf{L}^j\|_F\\
&\leq \|\mathbf{L}^j\|_{\max} \cdot (1 + O(\eta/n))\\
&\leq 2\cdot\|\mathbf{L}^j\|_{\max},
\end{aligned}$$

611 where the first inequality is triangle inequality, the second inequality follows by (36), the third
612 inequality follows from monotonicity of $L_p$ norms, the fourth inequality follows from (34), and the
613 fifth inequality follows from the fact that $\mathbf{L}^j$ is $s^2$ sparse.

614 Recalling (35) and the definition of $\Sigma_*^{\sigma,\tau;j}$, we conclude that $\|\Sigma_*^{\sigma,\tau;j}\|_{\max} \leq O(1)$ as claimed. $\qquad\square$

## C Sub-Exponential Tail Bounds From Section A

616 In this section, we provide proofs for Lemmas A.2, A.3, and A.4, restated here for convenience.

617 **Lemma A.2.** *Let $\xi > 0$ and let $\mathcal{N} \subset \mathbb{R}^{n\times n}$ be any finite set for which $\|\Sigma\|_{\max} \leq O(1)$ for all*
618 $\Sigma \in \mathcal{N}$. *Let $\mu_1,...,\mu_N,\overline{\mu} \in \Delta^n$ satisfy $\overline{\mu} \triangleq \frac{1}{N}\sum_{i=1}^N \mu_i$. Then for $X_i \sim Mul_k(\mu_i)$ for $i \in [N]$,*

$$\Pr\left[\left|\left\langle \frac{1}{N}\sum_{i=1}^N (X_i - \mu_i)^{\otimes 2} - \mathop{\mathbb{E}}_{X\sim Mul_k(\mu_i)}\left[(X-\mu_i)^{\otimes 2}\right], \Sigma\right\rangle\right| > t \,\forall\, \Sigma \in \mathcal{N}\right] < 2|\mathcal{N}|\exp\left(-\Omega\left(\frac{Nk^2t^2}{1+kt}\right)\right),$$

619 *where the probability is over the samples $X_1,\cdots,X_N$.*

620 **Lemma A.3.** *Let $\xi > 0$ and let $\mathcal{N} \subset \mathbb{R}^{n\times n}$ be any finite set for which $\|\Sigma\|_{\max} \leq O(1)$ for all*
621 $\Sigma \in \mathcal{N}$. *For $X_i \sim Mul_k(\mu_i)$ for $i \in [N]$, $\hat{\mu} \triangleq \frac{1}{N}\sum_{i=1}^N X_i$, and $\overline{\mu} \triangleq \frac{1}{N}\sum_{i=1}^N \mu_i$,*

$$\Pr\left[\left|\langle(\hat{\mu}-\mu)^{\otimes 2}, \Sigma\rangle - \mathbb{E}\left[\langle(\hat{\mu}-\mu)^{\otimes 2}, \Sigma\rangle\right]\right| > t \,\forall\, \Sigma \in \mathcal{N}\right] < 2|\mathcal{N}|\exp\left(-\Omega\left(\frac{N^2k^2t^2}{1+Nkt}\right)\right),$$

622 *where the probability is over the samples $X_1,\cdots,X_N$.*

623 **Lemma A.4.** *Let $\xi > 0$ and let $\mathcal{N} \subset \mathbb{R}^{n\times n}$ be any finite set for which $\|\Sigma\|_{\max} \leq O(1)$ for all*
624 $\Sigma \in \mathcal{N}$. *Let $\mu_1,...,\mu_N,\overline{\mu} \in \Delta^n$ satisfy $\|\mu_i - \overline{\mu}\|_1 \leq \omega$ for all $i \in [N]$. For $X_i \sim Mul_k(\mu_i)$ for*
625 $i \in [N]$,

$$\Pr\left[\left|\frac{1}{N}\sum_{i=1}^N (\mu_i - \overline{\mu})^\top \Sigma(X_i - \mu_i)\right| > \omega\cdot t \,\forall\, \Sigma \in \mathcal{N}\right] < 2|\mathcal{N}|\exp\left(-\Omega\left(kNt^2\right)\right),$$

626 *where the probability is over the samples $X_1,\cdots,X_N$.*

We remark that if we restricted our attention to test matrices of the form $\Sigma = vv^\top$ for $v \in \{\pm 1\}^n$, these lemmas would follow straightforwardly from Bernstein's and the sub-Gaussianity of binomial distributions.

We will need the following well-known combinatorial fact, a proof of which we include for completeness in Section C.1

**Fact C.1.** *For any $m, r \in \mathbb{Z}$, there are at most $O(m)^r \cdot r!$ tuples $(i_1, ..., i_{2r}) \in [m]^t$ for which every element of $[m]$ occurs an even (possibly zero) number of times.*

Central to the proofs of Lemmas A.2 and A.3 is the following sub-exponential moment bound. We remark that this moment bound would be an immediate consequence of McDiarmid's if $\Sigma$ not only satisfied $\|\Sigma\|_{\max}$ but was also psd, but because the matrices arising from shelling need not be psd, it turns out to be unavoidable that we must prove this moment bound from scratch.

In this section, given $\mu \in \Delta^n$, let $\mathcal{D}_\mu$ denote the distribution over standard basis vectors $\{e_i\}$ of $\mathbb{R}^n$ where for any $i \in [n]$, $e_i$ has probability mass equal to the $i$-th entry of $\mu$.

**Lemma C.2.** *Let $\Sigma \in \mathbb{R}^{n \times n}$ have entries bounded in absolute value by $O(1)$, and for $\mu_1, ..., \mu_m, \overline{\mu} \in \Delta^n$, let $\overline{\mu} \triangleq \frac{1}{m} \sum_{i=1}^m \mu_i$. If $Y_1, ..., Y_m$ are independent draws from $\mathcal{D}_{\mu_i}$ respectively, and $\hat{\mu} \triangleq \frac{1}{m} \sum_{i=1}^m Y_i$, then for every $r \geq 1$, $\mathbb{E}\left[\left((\hat{\mu} - \overline{\mu})^\top \Sigma (\hat{\mu} - \mu)\right)^r\right] \leq \Omega(m)^{-r} \cdot r!$.*

*Proof.* Without loss of generality, suppose $\Sigma$ has entries bounded in absolute value by 1. For $i, i' \in [m]$, define $Z_{i,i'} \triangleq (Y_i - \mu_i)^\top \Sigma (Y_{i'} - \mu_{i'})$. Note that because $\|Y_i - \mu_i\|_1 \leq 2$ with probability 1 for all $i \in [m]$, and the entries of $\Sigma$ are bounded in absolute value by 1, $|Z_{i,i'}| \leq 4$ with probability 1 for all $i, i' \in [m]$. We can write $\mathbb{E}\left[\left((\hat{\mu} - \overline{\mu})^\top \Sigma (\hat{\mu} - \overline{\mu})\right)^r\right]$ as

$$\frac{1}{m^{2r}} \mathbb{E}\left[\left(\sum_{i,i' \in [m]} Z_{i,i'}\right)^r\right] = \frac{1}{m^{2r}} \sum_{(i_1,i_1'),...,(i_r,i_r')} \mathbb{E}\left[\prod_{j=1}^r Z_{i_j,i_j'}\right]. \tag{37}$$

Now that if there exists some index $i \in [m]$ which occurs an odd number of times among $i_1, i_1', ..., i_r, i_r'$, then by the fact that the tensor $\mathbb{E}\left[(Y_i - \mu_i)^{\otimes a}\right]$ is identically zero for odd $a$, we have that $\mathbb{E}\left[\prod_{j=1}^r Z_{i_j,i_j'}\right]$. So the nonzero summands on the right-hand side of (37) correspond to indices $\{(i_j, i_j')\}_{j \in [r]}$ which must satisfy that every index appearing among $i_1, i_1', ..., i_r, i_r'$ appears an even number of times. By Fact C.1, there are $O(m)^r \cdot r!$ such tuples.

Finally, by the fact that $|Z_{i,i'}| \leq 4$ with probability 1 for all $i, i' \in [M]$, each monomial $\mathbb{E}\left[\prod_{j=1}^r Z_{i_j,i_j'}\right]$ is upper bounded by $4^r$. We conclude that $\mathbb{E}\left[\left((\hat{\mu} - \overline{\mu})^\top \Sigma (\hat{\mu} - \overline{\mu})\right)^r\right] \leq \frac{1}{m^{2r}} \cdot O(m)^r \cdot r! \cdot 4^r$, from which the claim follows. $\qquad\square$

Similarly, a crucial ingredient to the proof of Lemma A.4 is the following moment bound.

**Lemma C.3.** *Let $\Sigma \in \mathbb{R}^{n \times n}$ have entries bounded in absolute value by $O(1)$, and suppose $\mu_1, ..., \mu_m, \overline{\mu} \in \Delta^n$ satisfy $\|\mu_i - \mu_m\|_1 \leq \omega$ for all $i \in [m]$. Then for every $r \in \mathbb{Z}$, $\mathbb{E}\left[\left(\frac{1}{m} \sum_{i=1}^m (\mu_i - \overline{\mu})^\top \Sigma (Y_i - \mu_i)\right)^r\right]$ is 0 if $r$ is odd and at most $O(r\omega^2/m)^{r/2}$ otherwise.*

*Proof.* It is clear that the $r$-th moment is zero when $r$ is odd. Henceforth, write $r$ as $2r$. Without loss of generality, suppose $\Sigma$ has entries bounded in absolute value by 1. For $i \in [m]$, define $Z_i \triangleq (\mu_i - \overline{\mu})^\top \Sigma (Y_i - \mu_i)$. Note that because $\|Y_i - \mu_i\|_1 \leq 2$ with probability 1 for all $i \in [m]$, and the entries of $\Sigma$ are bounded in absolute value by 1, $|Z_i| \leq 2\omega$ with probability 1 for all $i \in [m]$. We can write $\mathbb{E}\left[\left(\frac{1}{m} \sum_{i=1}^m (\mu_i - \overline{\mu})^\top \Sigma (Y_i - \mu_i)\right)^{2r}\right]$ as

$$\frac{1}{m^r} \mathbb{E}\left[\left(\sum_{i \in [m]} Z_i\right)^r\right] = \frac{1}{m^r} \sum_{i_1,...,i_{2r}} \mathbb{E}\left[\prod_{j=1}^{2r} Z_{i_j}\right].$$

As in the proof of Lemma C.2, the only nonzero summands correspond to tuples $(i_1, ..., i_{2r})$ such that every element of $[m]$ appears an even (possibly zero) number of times. By Fact C.1, there are at most $O(m)^r \cdot r!$ such tuples, from which we can complete the proof. $\qquad\square$

Lemmas A.2 and A.3 will now follow as consequences of Lemma C.2 and the following standard tail bound for random variables with sub-exponential moments:

**Fact C.4.** *Let $Z_1, ..., Z_m$ be random variables for which there exists a constant $\nu > 0$ such that $\mathbb{E}[Z_i^r] \le \frac{1}{2}\nu^r \cdot r!$ for all integers $r \ge 1$ and $i \in [m]$. Then*

$$\Pr\left[\left|\frac{1}{m}\sum_{i=1}^m Z_i - \mathbb{E}[Z]\right| > t\right] \le 2e^{-\Omega\left(\frac{mt^2}{\nu^2+\nu t}\right)}.$$

Similarly, Lemma A.4 will follow as a consequence of Lemma C.3 and the following standard tail bound for random variables with sub-Gaussian moments:

**Fact C.5.** *Let $Z_1, ..., Z_m$ be random variables for which there exists a constant $\nu > 0$ such that $\mathbb{E}[Z_i^r] \le (r \cdot \nu^2)^{r/2}$ for all integers $r \ge 1$ and $i \in [m]$. Then*

$$\Pr\left[\left|\frac{1}{m}\sum_{i=1}^m Z_i - \mathbb{E}[Z]\right| > t\right] \le 2e^{-\Omega\left(mt^2/\nu^2\right)}.$$

*Proof of Lemma A.2.* This follows by taking $m = k$ in Lemma C.2 and $m = N$ in Fact C.4 and noting that for any $\Sigma \in \mathcal{N}$, $\|\Sigma\|_{\max} \le O(1)$ by Lemma A.1. $\qquad\square$

*Proof of Lemma A.3.* This follows by taking $m = kN$ in Lemma C.2 and $m = 1$ in Fact C.4 and noting that for any $\Sigma \in \mathcal{N}$, $\|\Sigma\|_{\max} \le O(1)$ by Lemma A.1. $\qquad\square$

*Proof of Lemma A.4.* This follows by taking $m = k$ in Lemma C.3 and $m = N$ in Fact C.5 and noting that for any $\Sigma \in \mathcal{N}$, $\|\Sigma\|_{\max} \le O(1)$ by Lemma A.1. $\qquad\square$

## C.1 Proof of Fact C.1

*Proof.* To count the number $N^*$ of such tuples $(i_1, ..., i_{2r})$, for every $1 \le s \le r$ let $N_s$ denote the number of tuples $\beta \in \{2, 4..., 2r\}^s$ for which $\sum_{i=1}^s \beta_i = 2r$. By balls-and-bins, $N_s = \binom{r+s-1}{r} \le \left(\frac{3es}{2r}\right)^r$. Now note that to enumerate $N^*$, we can 1) choose the number $1 \le s \le \min(m, r)$ of unique indices among $\{i_j\}$, 2) choose a subset $S$ of $[m]$ of size $s$, 3) choose one of the $N_s$ tuples $\beta$, and 4) choose one of the $\binom{2r}{\beta_1, ..., \beta_s}$ ways of assigning index $S_1$ to $\beta_1$ indices in $\{i_j\}$, $S_2$ to $\beta_2$ indices, etc. For convenience, let $r' \triangleq \min(m, r)$. We get an upper bound of

$$\begin{aligned}
N^* &\le \sum_{s=1}^{\min(m,r)} \binom{m}{s} \cdot N_s \cdot \binom{2r}{\beta_1, ..., \beta_s} \\
&\le \sum_{s=1}^{\min(m,r)} \frac{m^s}{s!} \left(\frac{3es}{2r}\right)^r \cdot (2s)! \\
&\le \frac{m^{r'}}{(r')!} \cdot r' \cdot \left(\frac{3er'}{2r}\right)^r \cdot (2r')! \\
&\le \frac{m^r}{(r)!} \cdot r \cdot (3e/2)^r \cdot (2r)! \\
&= m^r \cdot r \cdot (3e/2)^r \cdot \binom{2r}{r} \cdot r! \\
&\le O(m)^r \cdot r!,
\end{aligned}$$

where in the second step we used basic bounds on binomial and multinomial coefficients together with the above bound on $N_s$, in the third step we used the fact that the summands are increasing in $s$, and in the fourth step we used this fact along with the fact that $r' \le r$ by definition. $\qquad\square$



[Supplementary Material 2 · main_unstructured.pdf]



(i) domain size $n$

(ii) batch size $k$

(iii) corruption $\varepsilon$

(iv) number of batches

Legend: $\varepsilon/\sqrt{k}$, filter, oracle, naive

$y$-axis: $A_{\ell/2}$ distance

[Supplementary Material 3]



(i) domain size $n$

(ii) batch size $k$

(iii) corruption $\varepsilon$

(iv) number of batches

Legend: $\varepsilon/\sqrt{k}$, filter, oracle, naive