[Reviews · NeurIPS 2020]

Review 1

Summary and Contributions: We are given N batches of samples, of which epsilon fraction of are corrupted, and the remaining batches are from distributions that are omega close to a ground truth mu. The goal is to estimate mu to total variance. This paper considers the special case of structured distributions, and designs algorithms that run in polynomial time, and require a number of batches that is at most quadratic factor of the best known lower bound. This paper is a combination of two prior works. One [JO], which provides a simple polynomial time algorithm with near optimal sample complexity, and another [CLM] which requires superpolynomial number of batches but also consider more general classes of structured distributions. In this paper the authors study the problem of structured distribution estimation considered in CLM and apply the tools of JO to obtain the sample complexity bounds. — Post Author Response — I am happy with the author’s clear and honest response. I have revised my score.

Strengths: The paper proposes algorithms that are polynomial in running time and data requirement unlike the previous works for structured distribution estimation from untrusted batches. The paper requires technical insights to extend JO to the case of structured distributions and is definitely non-obvious.

Weaknesses: While it requires some technical insights, I miss if there is any additional technical insights that are needed beyond combining the methods in CLM and JO. In particular, perhaps the technical difficulties in directly combining the two methods should be clarified better.

Correctness: Yes.

Clarity: Yes.

Relation to Prior Work: Yes.

Reproducibility: Yes

Additional Feedback:


Review 2

Summary and Contributions: This paper considers the setting of learning discrete distributions from untrusted batches that was introduced in Qiao and Valiant recently. Here there are N batches of k i.i.d. samples each drawn from a distribution \mu over the discrete domain [n], and an \eps fraction of the batches can be corrupted arbitrarily. The goal is to learn the distribution \mu efficiently. This is related to the work on robust estimation, except that here, one needs to get closeness in L_1 norm (as opposed to L_2). Recently, Jain and Orlitsky'19 and Chen Li Moitra '19 designed efficient algorithms to solve this problem. The main setting considered in this paper is when the distribution \mu is structured i.e., it is close to a piecewise polynomial function (few pieces) of small degree each. The main contribution is a new polynomial time algorithm based on the adaptation of the filtering approach by [JO19], that also works in this structured setting. The sample complexity dependency on many of the parameters is tight. These bounds have a dependence mostly on the number of pieces and degree, and only a logarithmic dependence on dimension. The paper also demonstrates the effectiveness of their algorithm with simulated data, where an \eps fraction of the batches are drawn from a different distribution (which moves mass from the high-probability items to the low-probability items).

Strengths: The theoretical contribution is solid: it gives improved guarantees in the structured setting that was explored in Chen-Li-Moitra'19. This is an improvement from quasi-polynomial time dependence to polynomial time dependence in terms of the structure parameters (# pieces, degree), and in the sample complexity bounds. There is also some technical contribution by extending the filtering framework in the recent robust estimation literature to what they call soft filtering, which avoids the need for rounding the SDP in each step.

Weaknesses: I think the main drawback is that the paper is quite incremental in terms of the techniques. It combines the filtering approach in Jain and Orlitsky'19, and the ideas in CKM'19 of using the Haar wavelet basis (where the representation is sparse). Putting these together is still quite technical, but perhaps not surprising. The model of Qiao and Valiant is interesting. But the notion of "structured distribution" in terms of piecewise polynomial approximations considered here is not as convincing. This seems more of theoretical interest. In this context, it would have been more interesting to see experiments based on real data (where there is some potential structure), as opposed to simulated data from some fixed number of pieces. Finally the experiments are run on small values of n ~ 64 and n~128. The SDP-based algorithm doesn't seem practical (it takes several minutes for n~128). For such small n, it is not clear that the sample complexity savings in terms of n are significant. But this is less important in my opinion, since the main contribution is theoretical.

Correctness: The claims seem correct, as much as I could verify.

Clarity: The paper is dense, and is very hard to read, apart from the introduction. E.g., the algorithm (in particular, the SDP) is not fully specified in the main body, and there are incorrect references to equations etc. The paper as written seems to be aimed at the experts.

Relation to Prior Work: Yes

Reproducibility: Yes

Additional Feedback: Overall I am on the fence about this paper. ------ Post Rebuttal ----------- The rebuttal helped with some of the concerns that I had, particularly in the new ideas needed to extend the JO19 approach to this setting. I'm happy to see the paper accepted. Comments: 1. It may be good to see if there are more practical algorithms, by avoiding the use an SDP solver (or using algorithms based on multiplicative weights). I would recommend that the authors use a better SDP solver like MOSEK. It can often solve instances with #vars, constraints of the order of several hundred, or even thousands in the same time. 2. About the writing: the semidefinite program specified in the main body; the only reference is in Definition 2.3 where it is mentioned in passing. There are various references that appear before they are defined (e.g., constraints 3 and 5 in section 2). I assume this is referring to the definition in Section 2.1 of the Appendix? 3. In the soft filtering framework, does avoiding rounding the SDP in each step give significant running time savings as well, for the unstructured setting compared to [JO19](or is it primarily of technical interest)? If it does reduce the running time, it will be good to emphasize that.


Review 3

Summary and Contributions: The paper solves an important problem in machine learning consisting of determining a distribution on {1,...,n} given a number of minibatches coming from a close distribution and a number of corrupted minibatches. The main contribution of the paper is to provide an algorithm which scales sublinearly in n, while staying competitive with state of the art concerning the statistical accuracy.

Strengths: The paper answers a key question in the field concerning the algorithmic complexity of solving the problem of robust learning discrete distributions. This contribution is very relevant to the ML community and more specifically the

Weaknesses: Practical relevance beyond the theoretical achievements should perhaps be given some more consideration.

Correctness: The proofs I have reviewed are correct.

Clarity: In my opinion, the paper is very clearly written and a pleasure to read.

Relation to Prior Work: Relation to prior work is clearly stated and the contribution of the paper as compared to the current state of the art is illuminating.

Reproducibility: Yes

Additional Feedback:

[Author Response · NeurIPS 2020]

We would like to thank all the reviewers for taking the time to understand our work, and for their helpful comments!

To Reviewers #1 and #2: We thank both for raising the important question of what the contributions of this work are
beyond merely combining CLM and JO. Recall that the primary technical innovation of our work was to show that we
can estimate skewness with an SDP and filter the data *without needing to round the SDP*. We clarify below that filtering
without rounding is not just for aesthetic considerations or runtime savings: rather, it is an essential workaround to a
fundamental complexity-theoretic barrier that arises in our setting but in neither CLM nor JO in isolation. Indeed, as we
explain below, with this barrier in mind it should actually be quite surprising that our algorithm works at all.

**Reminder of JO's approach**   Recall the key subroutine in JO was to estimate "skewness" of the weighted dataset,
i.e. to estimate $\max_{v \in \{0,1\}^n} |v^\top A v|$ for some data-dependent matrix $A$. And the seminal paper of Alon-Naor on
approximating the cut-norm gives an SDP *and a rounding scheme* that together produce $v$ whose objective value is
within a constant factor of optimal. We emphasize that having access to a rounded solution, that is, an actual *bitstring* $v$
as opposed to a psd matrix, is essential for JO to be able to follow the conventional filtering analysis.

**A hardness of approximation barrier**   As mentioned in our technical overview, instead of optimizing $|v^\top A v|$
over $v \in \{0,1\}^n$, we need to optimize $|v^\top A v|$ over $v \in \mathcal{V}_\ell^n$. This is a far more challenging optimization problem
and closely connected to things like sparse PCA. Indeed, if $A' := T^\top A T$ where $T$ is the matrix which maps $v$ to
$(0, v_2 - v_1, v_3 - v_2, ..., v_n - v_{n-1})$, then this problem becomes that of optimizing $w^\top A' w$ over $\ell$-sparse vectors
$w \in \{0, \pm 1\}^n$ whose nonzero entries are alternating in sign. Modulo the alternating sign condition, this is at least as
hard as densest $\ell$-subgraph. Notably, Manurangsi (STOC '17) showed that under the Exponential Time Hypothesis, it is
even impossible to efficiently achieve any sub-poly$(n)$ factor approximation for this problem.

**Our workaround**   Existing works on filtering for robust statistics all require filtering along univariate projections of
the data, but in the unavoidable absence of a rounding scheme in our setting, the only approach one can try is to just
"project" along the solution to our SDP relaxation of $\max_{v \in \mathcal{V}_\ell^n} |v^\top A v|$ at each step, and this is the approach we take.

**In light of the above barrier, the fact that our algorithm works should be quite striking: the hardness of ap-**
**proximation result suggests that the integrality gap of our SDP relaxation should be horrible. In particular, the**
**norm $\| \cdot \|_{\mathcal{K}}$ induced by this relaxation might be very distorted relative to the $\mathcal{A}_\ell$ norm we actually care about.**

The saving grace is that "regularity" (in the sense of Definition 3.2 in the body) still holds with respect to this distorted
norm, which, together with the careful design of the SDP (e.g. so that we still have estimates like Corollary 2.5 in the
supplement) allows us to deduce "soundness" (in the sense of Lemma 3.5 in the body) with respect to this norm.

In fact, not being able to round introduces complications even in the "easiest" parts of the traditional analysis of the
filtering framework. For instance, even the statement that a large enough subset of the uncorrupted samples will satisfy
"$\epsilon$-goodness", which in JO followed from standard tail bounds for sub-exponential random variables, requires proving
new concentration inequalities (we explain this in Appendix C). The proof of soundness becomes similarly delicate.

Unrelatedly, another innovation over CLM is that, to achieve $O(\ell^2)$ sample complexity, we needed to significantly
tighten their metric entropy bound: even after implementing the above workaround, CLM's bound would have gotten
some impractical polynomial dependence on $\ell$. As alluded to in Remark 2.4 of the body, in order to get the guarantees
of Lemma 3.4 in the body, it was necessary to modify the convex relaxation they considered.

To Reviewer #2:

1. "E.g., the algorithm (in particular, the SDP) is not fully specified in the main body." The full SDP is given in
Definition 2.3 in the body, and the references to specific constraints in section 2 are references to the constraints in
Definition 2.3. The full algorithm is then given in Algorithm 1 and 2.

2. "But the notion of "structured distribution" in terms of piecewise polynomial approximations considered here is
not as convincing." The notion of piecewise polynomials (or splines) we work with is a standard one that has been
studied extensively in statistics, see the references in Section 1.2 of the main body. It strictly subsumes classes of
"structured" univariate distributions like monotone, multi-modal, concave, convex, log-concave, Gaussian, Poisson,
binomial, monotone hazard rate, Besov, etc., and arbitrary mixtures of such distributions.

3. The SDP-based algorithm doesn't seem practical (it takes several minutes for $n = 128$). For such small $n$, it is not
clear that the sample complexity savings in terms of $n$ are significant. Note that it is still possible to see the expected
factor of $n/\ell$ in sample complexity savings. For reference, in JO (see e.g. their Figure 1), for alphabet size 100 and
corruption fraction 0.4, they need to draw $100/(0.4)^2 = 625$ batches. In contrast, for $\ell = 10$ and $n = 128$, we only
draw $\ell/(0.4)^2 \approx 63$ batches. That said, we completely agree that getting a more practical non-SDP-based algorithm
that can work for much larger $n$ is an important direction for future work.

[Meta-Review · NeurIPS 2020]

This paper addresses the question of learning structured distributions from batches when a constant fraction of the batches might be corrupted. This problem has been of considerable recent interest. This paper studies the setting where the underlying distribution has additional structure (namely, piece polynomial density function), in which case more sample efficient algorithms are possible. This paper develops sample and computationally efficient algorithms for such settings. The reviewers were convinced that this paper makes important technical contributions in extending recent work on this problem to the structured setting. I recommend accepting this paper.